METHODS

# Putting BASIL in a BLT: A Bayesian filtering method for estimating the fitness effects of nascent adaptive mutations

Huan-Yu Kuo[1,2], Sergey Kryazhimskiy[1]*

**1** Department of Ecology, Behavior and Evolution, University of California San Diego, La Jolla, California, United States of America, **2** Department of Physics, University of California San Diego, La Jolla, California, United States of America

* skryazhi@ucsd.edu

## Abstract

The distribution of fitness effects (DFE) of new beneficial mutations is a key quantity that dictates the dynamics of adaptation. The barcode lineage tracking (BLT) approach is an important advance toward measuring DFEs. BLT experiments enable researchers to track the frequencies of ~$10^5$ barcoded lineages in large microbial populations and detect up to thousands of nascent beneficial mutations in a single experiment. However, reliably identifying adapted lineages and estimating the fitness effects of driver mutations remains a challenge because lineage dynamics are subject to demographic and measurement noise and competition with other lineages. We show that the commonly used Levy-Blundell method for analyzing BLT data and its improved version FitMut2 can produce biased fitness estimates, particularly if selection is strong. To address this problem, we develop a new method called BASIL (BAyesian Selection Inference for Lineage tracking data), which dynamically updates the belief distribution of each lineage's fitness and size based on the number of barcode reads. We calibrate BASIL's model of noise with new experimental data and find that noise variance scales non-linearly with lineage abundance. We test how BASIL and FitMut2 perform on simulated data and on down-sampled data from the original BLT data by Levy et al and find that BASIL is both more robust and more accurate than FitMut2. Our work paves the way for a systematic inference of the distribution of fitness effects of new beneficial mutations from BLT experiments in a variety of scenarios.

## Author summary

Beneficial mutations are rare but they are the ultimate drivers of evolution by natural selection. Evolutionary biologists seek to understand how many beneficial mutations an organism has access to in different environments and how these

**Data availability statement:** Code and data availability Code. All analysis code is written in Python and is available at https://github.com/HuanyuKuo/BASIL-public. Computationally intensive analyses were run on the Triton Supercomputing Cluster (TSCC). Data. Barcode sequencing data generated in this work are available at https://www.ncbi.nlm.nih.gov/bioproject/1227420.

**Funding:** This work was supported by the Career Award at Scientific Interface (1010719.01) from the Burroughs Wellcome Fund to SK, the Alfred P. Sloan Research Fellowship (FG-2017-9227) to SK, the Hellman Fellowship to SK, and the NIH grants 1R01GM137112 and R35GM153242 to SK. SK salary was supplemented by NIH, HK salary was supplemented by all funders. The funders had no role in study design, data collection and analysis, decision to publish, or preparation of the manuscript.

**Competing interests:** The authors have declared that no competing interests exist.

mutations affect fitness. Barcode lineage tracking (BLT) is a powerful experimental approach that tracks the frequencies of hundreds of thousands of subpopulations labeled with unique DNA barcodes and provides data that potentially enables researchers to identify and isolate many beneficial mutations arising in experimental microbial populations. However, analyzing these data is challenging because of the randomness of evolution and measurement noise. We found that existing methods for analyzing BLT data can lead to biased estimates of the fitness effects of beneficial mutations, especially when selection is strong. To overcome this issue, we developed a new method called BASIL, which uses a Bayesian approach that updates the estimated fitness and size of each lineage based on the measured barcode counts. We show that BASIL provides more accurate and robust estimates of the fitness effects of beneficial mutations in both simulated and real datasets than the existing alternatives. Thus, BASIL will facilitate a better understanding of beneficial mutations and adaptation more generally.

## Introduction

Every new adaptation originates in a single individual. Most vanish soon thereafter. The few that survive may encounter fierce competition with other contending adapted lineages [1–5]. This competition—termed "clonal interference"—is particularly severe in organisms that have limited recombination and also have a large supply of beneficial mutations, such as many bacteria [6–8], cancers [9,10], and viruses [11–13]. In the clonal interference regime, the chance of fixation of a new allele depends not only on its own fitness benefit but also on the benefits provided by competing alleles [14–16]. Thus, to understand the dynamics of rapid adaptation, we must know the distribution of fitness effects (DFE) of adaptive mutations [16,17]. However, this empirical knowledge is not yet readily available.

Measuring the effects of new adaptive mutations is challenging. The most statistically sound strategy would be to isolate them from random samples of spontaneous mutations. However, in most organisms, mutations are rare [18], and beneficial mutations typically constitute a relatively small fraction of all mutations [3,19,20] (but see [21,22]). As a result, obtaining accurate statistics of beneficial mutations from samples of random mutations is generally extremely inefficient. Instead, to sample adaptive mutations, researchers leverage the power of natural selection, which elevates the frequencies of beneficial mutations in a population. It is relatively easy to find and isolate beneficial mutations from recently adapted populations, particularly those evolved in the lab [23–26]. However, this strategy preferentially captures mutations with the strongest fitness benefits because they are least likely to be lost by genetic drift or clonal interference. Therefore, this approach may provide an incomplete view of the diversity of adaptive mutations [3]. Moreover, this view may also be biased because which mutations survive and fix depends in a complex way on the DFE itself and on population size and structure [16, 27, 28].

The barcode lineage tracking (BLT) method is a powerful strategy for estimating the fitness effects (also referred to as selection coefficients) of many nascent beneficial mutations in asexual genetically tractable microbes [3–5,29–36]. The key idea underlying BLT is to introduce many (typically ~$10^5$) random neutral DNA barcodes into an otherwise isogenic population and track barcode frequencies over time using deep sequencing [3,37]. Each new adaptive mutation that arises in the population is permanently linked to a unique DNA barcode. As an adaptive mutation spreads in the population, the corresponding increase in the frequency of the linked barcode can be detected in the sequencing data. Moreover, since the sizes of most barcoded subpopulations are initially small (typically ~100 individuals), the expansion of each adapted subpopulation is initially driven by a single adaptive mutation. In other words, for a period of time, each barcode with increasing frequency reports on a single adaptive driver mutation. As a result, this approach allows one to measure the fitness benefits of many simultaneously segregating driver mutations, including weak ones, in a single relatively short evolution experiment, while avoiding laborious genetic reconstructions and screens [3,38]. Although the BLT approach does not entirely eliminate biases introduced by natural selection, it significantly mitigates them, and overall provides a much more accurate picture of the diversity of adaptive mutations than other existing methods [3].

While the BLT approach significantly advances our ability to capture and study adaptive mutations, analyzing BLT data remains difficult. The main challenge is to distinguish lineages that have acquired a single beneficial mutation ("adapted lineages") from those that have not ("neutral lineages") based on noisy dynamics of low-abundance barcodes. The core idea is the same as in classical competition assays [39–42], i.e., to determine whether a lineage systematically increases in frequency relative to neutral reference lineages (i.e., those with wildtype fitness) and then to infer the fitness effect of the underlying driver mutation from the rate of this increase. However, despite many similarities, the BLT setup differs in three important ways from the canonical competition assay. First, in competition assays, lineages are usually represented by thousands of cells at the beginning of the experiment, whereas in the BLT experiments all lineages are by design present at low abundances (~100 cells), which makes lineage extinctions much more likely. Second, competition assays are typically short (≤ 50 generations) to prevent new mutations from significantly affecting lineage dynamics, whereas BLT experiments are longer (> 100 generations) precisely to allow for new adaptive mutations to arise and reach sufficiently high frequencies. Third, in competition assays, designated neutral reference lineages are deliberately added to the population. Their frequency dynamics reports on population's mean fitness which allows one to estimate the fitness of all other lineages relative to the reference [22,41]. In contrast, in the BLT setup, all lineages are initially equivalent, and since adaptive mutations can arise during the experiment (or shortly before), which lineages become adapted and which ones remain neutral is a priori unknown.

The absence of defined neutral reference lineages is perhaps the biggest challenge of analyzing BLT data because such lineages provide the most obvious and robust way to readout the mean fitness of the population, which in turn is required for the inference of fitness of all other lineages. Two main approaches addressing this problem have been proposed so far. Levy et al developed the original approach, which we refer to as the "Levy-Blundell" or the "LB" method for short [3]. They chose low-abundance lineages as the neutral reference, reasoning that such lineages are unlikely to acquire adaptive mutations during the BLT experiment, and inferred mean fitness based on the rate of decline in the frequency of such lineages. However, low-abundance lineages are likely to go extinct, especially if selection is strong. One could choose more abundant lineages as the neutral reference, but such lineages are more likely to carry adaptive mutations, especially later in the experiment. Both outcomes are undesirable as they can bias the estimates of fitness effects of all adaptive mutations. More recently, Li et al developed an approach called FitMut2, which retains many features of the original LB method but instead of relying on designated neutral reference lineages it iteratively switches between identifying adapted lineages and estimating mean fitness on their basis until the process converges [43]. They have shown that FitMut2 outperforms the original LB approach on their simulated data. However, FitMut2 could also be prone to biases because the solution to which it converges might be sensitive to initial conditions and/or noise in the data.

Given these potential theoretical concerns, it is important to assess the performance of existing approaches empirically on simulated data where the ground truth is known, and to understand when and why they fail. In particular, the ability to accurately infer fitness effects will likely depend on the strength of selection because stronger selection accelerates evolutionary dynamics and reduces the amount of time available for sampling lineage trajectories. To this end, in the first part of this paper, we assess the performance of the LB and FitMut2 approaches on BLT data simulated under weak and strong selection. We find that both approaches perform well under weak selection but produce substantially biased fitness estimates under strong selection. Since the LB method has been more widely used [3,4,29,31,32,35], we carry out a deeper investigation of the underlying reasons for its poor performance.

In the latter part of the paper, we develop a new BLT analysis method termed BASIL (BAyesian Selection Inference for Lineage tracking data), which keeps track of belief distributions for the fitness of all lineages under the assumption that most lineages are initially neutral. It updates these distributions based on barcode read counts, and uses random lineages (both putatively adapted and neutral) to estimate mean fitness. Our approach combines several techniques developed previously for the analyses of competition assays [42,44–46] and BLT experiments [3,5,43]. In particular, similar to previous methods, BASIL models the evolutionary dynamics of lineages between sampling time points as well as the measurement process, whose properties we measure using a new calibration experiment. In contrast to most other approaches and similar to Ref. [45], BASIL treats unobserved lineage sizes as hidden variables, which is important for obtaining accurate estimates of lineage fitness, particularly under strong selection. Similar to Ref. [42], we cast our model in the Bayesian framework, which allows us to keep track of uncertainties in our fitness and lineage size estimates and dynamically update our belief distributions as the data arrives. We then directly compare the performance of BASIL on published BLT datasets to the performance of FitMut2. Finally, we use BASIL to characterize adaptation of several strains of yeast *Saccharomyces cerevisiae* to various environments.

## Results

### Existing BLT analysis methods can produce biased fitness estimates

To understand how accurately the LB and FitMut2 methods infer the fitness effects of nascent adaptive mutations, we simulate a batch-culture BLT experiment with $10^5$ barcoded lineages, mimicking the setup of Levy et al [3]. The details of our simulations are provided in Materials and Methods. Briefly, each batch-culture cycle begins with dilution, which we simulate by down-sampling each lineage to $1/D = 1/256$ of its size using the Poisson distribution. Some lineages may go extinct at this step. Then, all surviving lineages deterministically expand by a factor proportional to the difference between their fitness and the population's mean fitness, so that the population reaches approximately its pre-dilution size. Our simulations continue for 20 cycles, corresponding to 160 generations. Every other cycle, we simulate barcode sequencing by randomly sampling a certain number of individuals from our population and recording their lineage identities.

In our simulations, 3,000 lineages are adapted, i.e., all individuals within these lineages have the same fitness advantage compared to the rest of the population. This scenario where all beneficial mutations are pre-existing is less complex than many real BLT experiments, and inference methods should perform best in this case. Our reasoning for using these relatively simple simulations was twofold. First, many if not most adaptive mutations detected in BLT experiments arise prior to the beginning of the experiment [3,4,33]. Second, we reasoned that if we find that a method fails on these relatively simple simulated data, it is very unlikely to perform well on real BLT datasets. Since there are no new mutations, our simulations are similar to competition assays, with the exception that the identity of neutral and adapted lineages is unknown to the inference algorithm. We simulate two conditions that we refer to as "strong selection" and "weak selection" (S1 Fig and S3 Table). In the weak selection regime, which resembles the original BLT experiment carried out by Levy et al [3], the average fitness of an adapted lineage is 3%. As a result, the mean fitness of the population increases by about 4% in 150 generations. In the strong selection regime, the average fitness of an adapted lineage is 8%, and the population's mean fitness increases by about 12% in 150 generations.

Our first goal is to assess how the LB and FitMut2 methods perform on these simulated data. If the methods underperform, we would like to understand why, focusing specifically on the better-established LB method. To enable a potential in-depth investigation, we simplify the original LB approach by stripping away much of the complexity required for analyzing real BLT datasets but unnecessary for the analysis of our simulated data, while retaining its core assumptions (see Materials and Methods and Section 2.3 in S1 Text). We refer to this simplified version as the "neutral decline" method to distinguish it from the original. As the original LB approach, the neutral decline method requires us to specify which lineages we select as the neutral reference. To determine how this choice might affect inference, we select either low- or high-abundance lineages, with low-abundance lineages being defined as those represented by 20–40 reads (corresponding to 20–40 cells at the bottleneck in our simulations) and high-abundance lineages being defined as those represented by 80–100 reads (corresponding to 80–100 cells at the bottleneck in our simulations).

The performance of both the neutral decline and FitMut2 methods on our BLT simulations are illustrated in Fig 1. We find that in the weak selection regime, both FitMut2 and the neutral decline method with high-abundance reference lineages accurately infer the mean fitness trajectory and the fitness effects of individual adaptive mutations (Fig 1E, 1F, 1I, and 1J). When low-abundance lineages are used as reference, the neutral decline approach underestimates both the mean fitness of the population and the selection coefficients of individual adaptive mutations even in this favorable regime (Fig 1A and 1B). Both methods perform significantly worse in the strong selection regime (Fig 1C, 1D, 1G, 1H, 1K, and 1L). FitMut2 and the neutral decline approach with high-abundance reference lineages severely underestimate both the mean fitness and the effects of individual mutations (Fig 1G and 1H). If low-abundance lineages are used as reference, the neutral decline method estimates the mean fitness accurately, but only for the first ~50 generations (Fig 1C), which results in noisy estimates of selection coefficients of individual adaptive mutations (Fig 1D).

Thus, the performance of existing BLT analysis methods depends on the conditions of the BLT experiment, in particular on the strength of selection. Furthermore, the accuracy of the neutral decline method—and by extension that of the LB method—hinges on the choice of reference lineages.

## Statistical causes for biases of the LB method

To understand why the neutral decline method (and, consequently, the LB method) sometimes produces biased estimates of lineage fitness, we first summarize the central assumption underlying this approach (see Materials and Methods and Table A in S1 Text for more details). To infer the population mean fitness $\bar{s}_k$ at the sampling interval $(t_{k-1}, t_k)$, the neutral decline method groups together reference lineages with each read count $r_{k-1}$ observed at $t_{k-1}$. For each such group, it computes the average read count $\bar{r}_k$ at the next sampling time $t_k$ and infers mean fitness $\bar{s}_k$ from the equation

$$\frac{\bar{r}_k}{R_k} = \frac{r_{k-1}}{R_{k-1}} e^{-\bar{s}_k \, \Delta t_k},$$

(1)

where $R_k$ is the total read depth at time $t_k$ and $\Delta t_k = t_k - t_{k-1}$. Equation (1) can be derived from the standard population genetics theory under the crucial assumption that all lineages with the same number of reads $r_{k-1}$ are present in the population at frequency $r_{k-1}/R_{k-1}$ at $t_{k-1}$ [47]. We refer to equation (1) as the "neutral decline" equation. Once the mean fitness trajectory $\bar{s} = (\bar{s}_1, \bar{s}_2, \ldots)$ is known, the neutral decline method infers the selection coefficients of all non-reference lineages by maximizing the likelihood of their frequency trajectories (for details, see Materials and Methods). Importantly, even though the original LB method is more sophisticated in that it utilizes the entire conditional distribution of lineage read counts at the next sampling time given their read counts at the previous time point (see equation (45) in the Supplementary material to Ref. [3] or equation (S8) in S1 Text), the mean of that distribution is given by equation (1). Therefore, a failure of equation (1) to capture the relationship between $r_{k-1}$ and $\bar{r}_k$ would imply a failure of the original LB model.

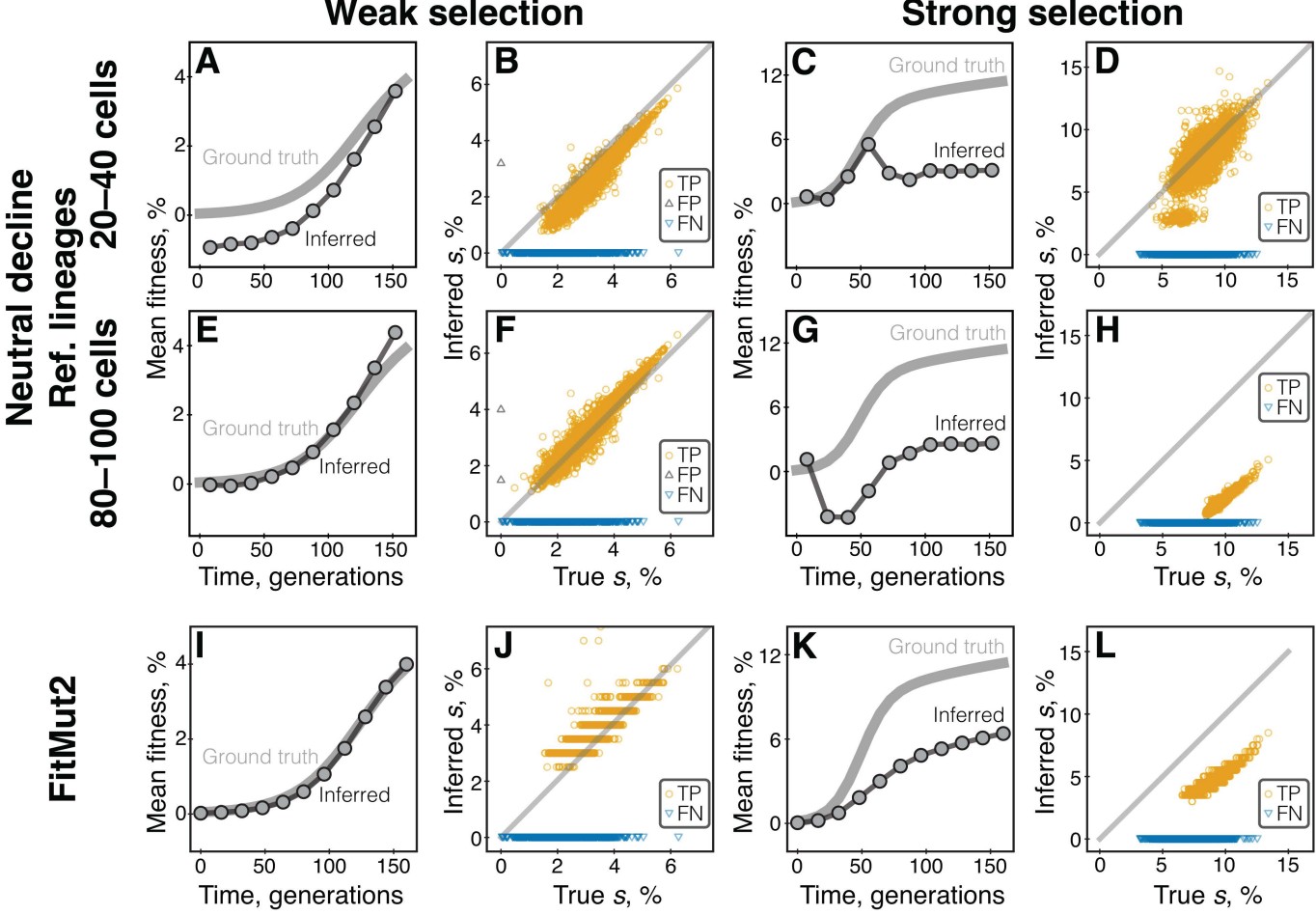

**Fig 1. Performance of the neutral decline and FitMut2 methods on simulated BLT data depends on the strength of selection.** Panels A–H show the results for the neutral decline method, and panels I–L show the FitMut2 results. Left panels A, B, E, F, I, J show simulations in the weak selection regime; right panels C, D, G, H, K, L show simulations in the strong selection regime. **A, C.** Mean fitness trajectories, true and inferred by the neutral decline method with low-abundance reference lineages. **B, D.** Selection coefficients of adapted lineages, true versus inferred by the neutral decline method with low-abundance reference lineages. Each symbol corresponds to a lineage; orange circles, black upward triangles and blue downward triangles represent true positives, false positives and false negatives, respectively; true negatives are not displayed for clarity. **E–H.** Same as A–D but for high-abundance lineages. **I–L.** Same as A–D but for FitMut2.

To test the validity of underline equation (1), we plot the rescaled average read count $\widetilde{r}_k = \bar{r}_k R_{k-1}/R_k$ at time point $t_k$ against the corresponding observed read count $r_{k-1}$ at the previous time point. According to equation (1), $\widetilde{r}_k$ must be proportional to $r_{k-1}$ with a zero $y$-intercept. Instead, we find that this relationship is more complex, with its shape being dependent on the selection regime and on time (Fig 2A and 2C). Specifically, when selection is weak, $\widetilde{r}_k$ relates linearly to $r_{k-1}$ during the entire BLT experiment, but with a non-zero $y$-intercept (Fig 2A). Despite its relatively small value (e.g., 6.60 at generation 16 and 2.89 at generation 160), a non-zero $y$-intercept can cause significant underestimates of mean fitness when low-abundance lineages are chosen as reference (Fig 2B), including negative estimates at early time points (see Fig 1A). Since the estimates of selection coefficients of all lineages depend on the estimate of mean fitness, the bias in the latter propagates to a bias in the former, as seen in Fig 1B. Choosing initially more abundant lineages as reference produces a more accurate estimate of initial mean fitness (Figs 1E and 2B) because such estimates are less sensitive to the value of

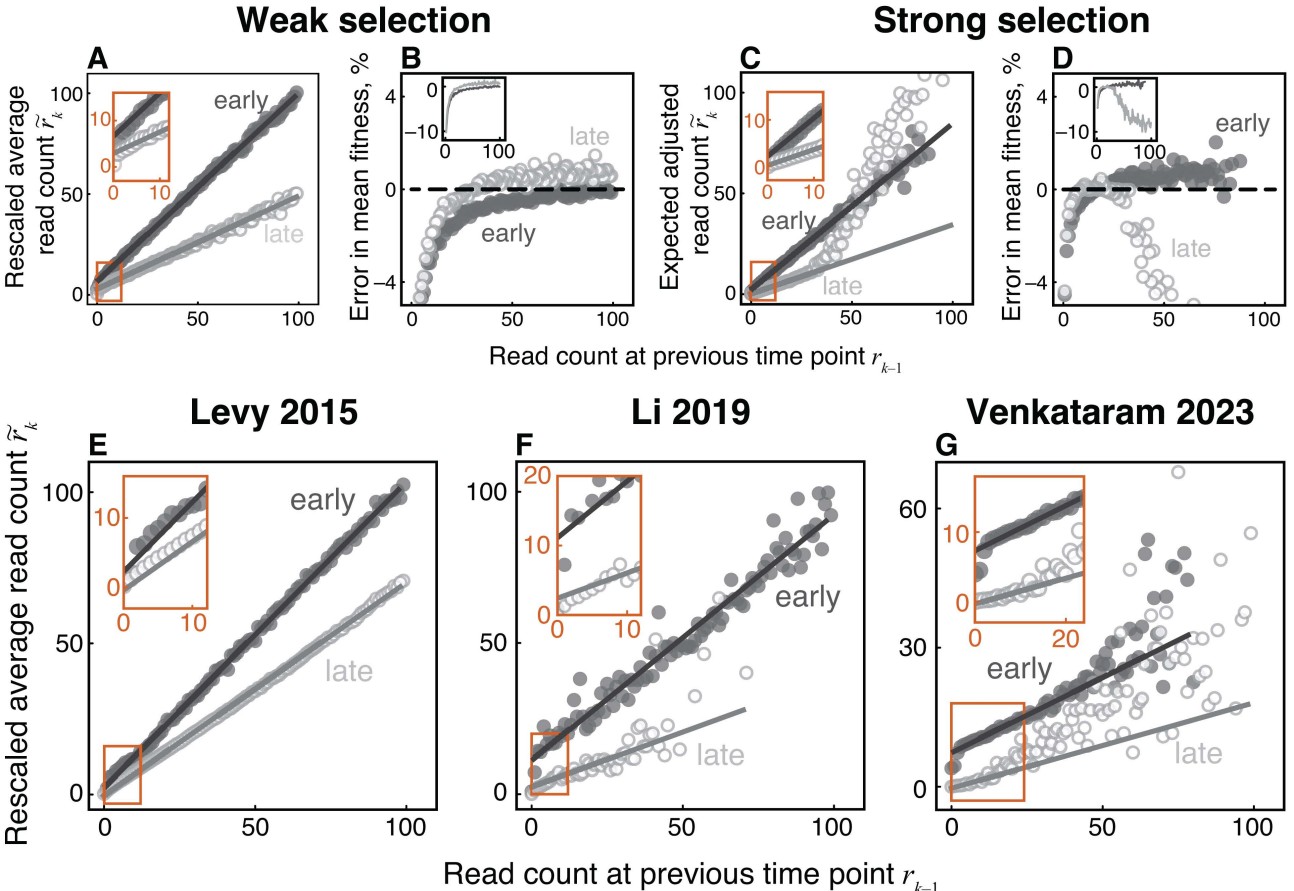

**Fig 2. Observed lineage read counts deviate from those predicted by equation (1) both in simulated and real data. A.** Rescaled average read count $\widetilde{r}_k = \bar{r}_k R_{k-1}/R_k$ at the next time point (ordinate) plotted against the read count observed at the previous time point $r_{k-1}$ (abscissa) for the weak-selection simulation. Light and dark gray points represent early ($t_k = 16$ generations) and late ($t_k = 160$ generations) time intervals. Lines represent least-squares best fits within the linear regime (see Materials and Methods). **B.** Error in the inferred mean fitness (inferred minus true) as a function of the read count of lineages that are used as the neutral reference. Shades are the same as in panel A. Early time interval is at $t_k = 16$ generations and late time interval is at $t_k = 64$ generations. **D.** Same as panel B but for the strong selection regime. **E–G.** Same as panel A but three real BLT datasets: Levy 2015 R1 (panel E), Li 2019 Evo1D R2 (panel F) and Venkataram 2023 Co-evolution R5 (panel G). Early time interval is at $t_k = 16, 7, 20$ generations and late time interval is at $t_k = 112, 133, 86$ generations in panels E, F, G, respectively.

the $y$-intercept (Fig 2A). However, choosing more abundant lineages is not universally better. For example, under strong selection, the relationship between $\widetilde{r}_k$ and $r_{k-1}$ becomes non-linear for higher-abundance lineages, particularly later in the experiment (Fig 2C), which also biases the estimates of mean fitness (Figs 1C, 1G and 2D).

These observations show that the relationship between lineage read counts at successive time points can be used to diagnose potential inference problems in the absence of the ground truth. Thus, we examined these relationships in real BLT data, asking whether they exhibit similar deviations from equation (1) as we observed in our simulations. To this end, we reanalyzed data from three published BLT studies, Levy 2015 [3], Li 2019 [29] and Venkataram 2023 [33]. We find strong deviations from linearity in the Li 2019 and Venkataram 2023 data for higher-abundance lineages at later time points (Fig 2F and 2G). Even at the initial time point where we expect equation (1) to be most accurate, we find that, although $\widetilde{r}_k$ depends on $r_{k-1}$ linearly, there is a statistically significant positive $y$-intercept (2.26 for Levy 2015, 11.10 for Li 2019, and 7.32 for Venkataram 2023, all $P$-values $< 10^{-12}$, F-test; Fig 2E–2G). As a result, the estimates of mean fitness

and selection coefficients of adaptive mutations in real data produced by the LB method are likely subject to the same biases as observed in simulated data sets.

Finally, we asked why the observed relationship between $r_{k-1}$ and $\bar{r}_k$ deviates from the prediction given by equation (1). First, the deviation from linearity at high $r_{k-1}$, especially later in the experiment (Fig 2C, 2F, and 2G), can be explained if reference lineages are in fact not neutral. Indeed, after selection had time to act, lineages that carry stronger beneficial mutations are typically present at higher frequencies in the population and hence have higher values of $r_{k-1}$ than neutral lineages. As a result, lineages with higher $r_{k-1}$ increase disproportionately more or decrease disproportionately less than lineages with lower $r_{k-1}$. Second, to understand why the relationship between $r_{k-1}$ and $\bar{r}_k$ has a non-zero $y$-intercept, consider a hypothetical example where all lineages are neutral and equally abundant in the population and the coverage at both time points $t_{k-1}$ and $t_k$ is such that a typical lineage is represented by 10 reads. Then, at $t_{k-1}$, while most lineages are represented by $r_{k-1} = 10$ reads, measurement noise will result in many lineages with $r_{k-1} = 9$, $r_{k-1} = 11$, $r_{k-1} = 8$, $r_{k-1} = 12$, etc. Since all lineages are neutral, the expected read count for any lineage at the next time point is $\bar{r}_k = 10$, including those with $r_{k-1} \neq 10$, whereas the neutral decline equation (1) predicts $\bar{r}_k = r_{k-1}$. This bias arises as a result of the regression to the mean, whereby lineages whose read counts are by chance abnormally low at one time point are represented by a more typical (larger) number of reads at the next time point, causing the $y$-intercept to be non-zero (see S1 Text, Section 2.3.2 for an extended discussion). The specific value of the intercept depends, among other things, on the measurement noise and on the distribution of lineage frequencies in the population, quantities that are a priori unknown. Therefore, correcting for this intercept appears difficult.

Overall, this investigation demonstrates that the neutral decline approach (and, consequently, the LB method) can lead to biased inferences of the fitness effects of nascent adaptive mutations because equation (1) (or the corresponding equation (S8) in S1 Text for the full distribution of reads) is in general incorrect, particularly for low-abundance lineages or under strong selection.

## BASIL: Bayesian selection inference for lineage tracking data

To overcome challenges of the existing approaches, we developed BASIL (Bayesian Selection Inference for Lineage tracking data), a robust statistical method for identifying adapted lineages and inferring their fitness and implemented it in a software package [48]. The key hidden variables in BASIL are the sizes $n$ and selection coefficients $s$ (fitness) of individual lineages. Population's mean fitness $\bar{s}_k$ and the measurement noise parameter $\epsilon_k$ defined below are time-dependent global variables. At any given sampling time $t_k$, each lineage $i$ is characterized by the joint belief distribution $P_{ik}^{\text{belief}}(n, s)$ for its size and fitness, which we model parametrically as a product of a normal and gamma distributions (see Materials and Methods and equations (S31)–(S33) in S1 Text). BASIL is a Bayesian filtering model (Fig 3B, [49]), in which the belief distributions $P_{ik}^{\text{belief}}(n, s)$ are based on the past data, i.e., data accumulated up to and including time $t_k$.

At BASIL's core is the model of evolutionary dynamics that computes the prior distribution $P_{ik}^{\text{prior}}(n, s)$ for the size and selection coefficient of each lineage $i$ at the current sampling time $t_k$ based on the past belief distribution $P_{ik-1}^{\text{belief}}(n, s)$ and the population's mean fitness $\bar{s}_k$ in the time interval $(t_{k-1}, t_k)$ (Fig 3A). We also develop a model of measurement $P^{\text{meas}}(r|n; \epsilon)$, which probabilistically relates the true lineage size $n$ to the corresponding barcode read count $r$ observed in the sequencing data. We model $P^{\text{meas}}(r|n; \epsilon)$ as a negative binomial distribution that depends on the current value of the noise parameter $\epsilon$.

Estimation of $P_{ik}^{\text{belief}}(n, s)$ at $t_k$ occurs in three steps (Fig 3C). First, we estimate the mean fitness of the population $\bar{s}_k$ and the noise parameter $\epsilon_k$ in the current time interval $(t_{k-1}, t_k)$ by sampling thousands of random lineages, computing their prior distributions $P_{ik}^{\text{prior}}(n, s)$ and then maximizing the likelihood of their observed read counts $r_{ik}$ at $t_k$. Second, we compute the prior distributions $P_{ik}^{\text{prior}}(n, s)$ for all lineages, given the estimated population's mean fitness $\bar{s}_k$. We then apply the Bayes theorem to obtain the updated lineage belief distributions $P_{ik}^{\text{belief}}(n, s)$, given the read counts $r_{ik}$ and the estimated noise parameter $\epsilon_k$. In the rest of the section, we flesh out the main ideas and mathematical expressions behind our method. The full details are provided in Section 3 in S1 Text.

# Bayesian Selection Inference for Lineage Tracking Data (BASIL)

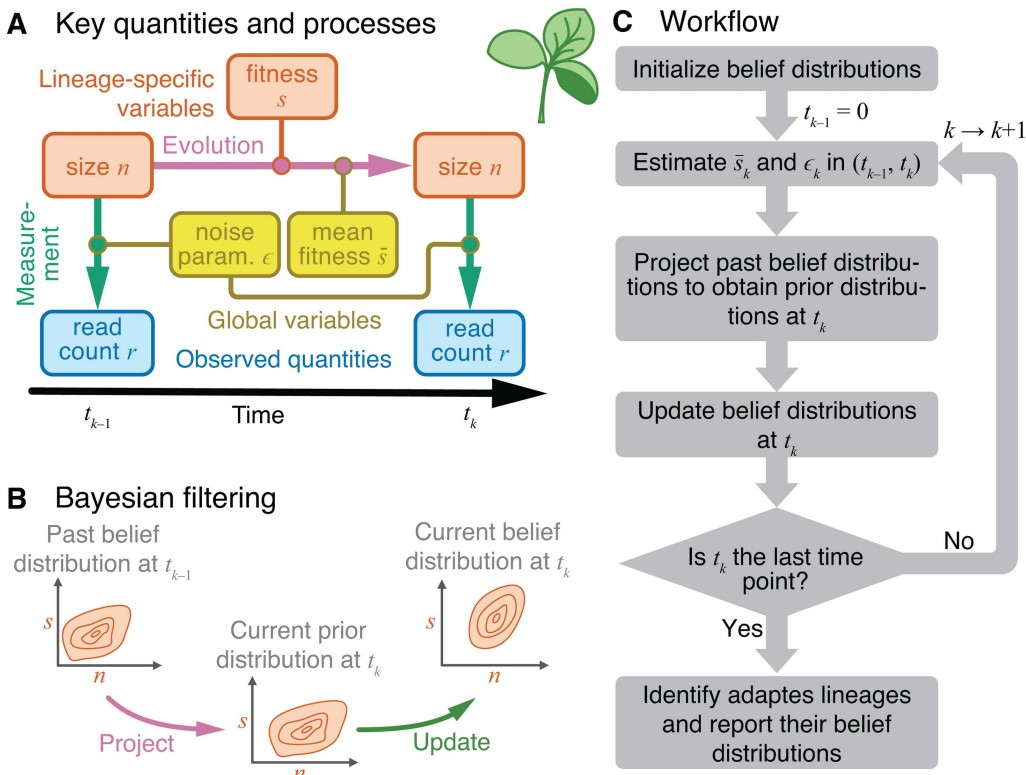

**Fig 3. BASIL schematic. A.** Key quantities and processes. Each barcoded lineage $i$ is characterized at the current time $t_k$ by its size $n_{ik}$ and fitness (selection coefficient) $s_{ik}$, which are unknown. Lineage size changes over time depending on the lineage fitness and the mean fitness of the population. At each sampling time point, we measure relative lineage sizes by counting sequencing reads with the corresponding barcode. **B.** At the first step of Bayesian filtering, we use the model of evolution to project the past belief distribution for the hidden variables $n$ and $s$ and obtain their current prior distribution. At the second step, we update this distribution based on the observed read count $r_k$. **C.** BASIL workflow.

**Model of evolution.** Consider a barcoded lineage with fitness $s$ relative to the ancestor and suppose that this lineage is represented by $n_{k-1} = n\left(t_{k-1}\right)$ individuals in the population at the previous sampling time $t_{k-1}$. On average, such lineage will expand if $s$ is larger than the population's current mean $\bar{s}$ and it will contract if $s < \bar{s}$. Demographic noise will also cause its size to randomly fluctuate around this expectation, such that the lineage size $n_k$ at the current sampling time $t_k$ is a random variable. To model it, we assume that our BLT population grows in the batch-culture regime [25] and sampling happens immediately prior to a dilution step. Then, during dilution, a fraction $1/D$ of the population is transferred into the fresh medium and the rest is discarded, where $D > 1$ is the dilution factor. Immediately after dilution, the size of the focal lineage becomes $\widetilde{n}$, which we draw from the Poisson distribution with mean $n_{k-1}/D$. Then, we assume that the lineage grows or shrinks exponentially and deterministically for the batch-culture cycle duration $\Delta t_c$, such that by the end of the cycle its size becomes $n\left(t_{k-1} + \Delta t_c\right) = \widetilde{n}De^{(s-\bar{s})\Delta t_c}$. If the next sampling point is the end of this cycle, then $n_k = n\left(t_{k-1} + \Delta t_c\right)$. Otherwise, we repeat the dilution and growth phases until the sampling time $t_k$ is reached. Note that this model allows for the possibility of lineage extinction during dilution.

**Model of measurement.** The actual lineage sizes are not observable. Instead, the population is sampled at time points $t_0$, $t_1$, … and sequenced at the barcode locus, which involves a series of processing steps [50]. Then the sequenced reads containing each barcode are counted. This measurement process can be described by a probability distribution

$P^{\text{meas}}(r|n)$ of observing $r$ reads for a barcode lineage with $n$ cells. The simplest model for this process is the Poisson distribution with mean $nR/N$, where $R$ is the read depth (i.e., the total number of reads obtained for the sample) and $N$ is the total population size at the sampling point. However, sequencing read counts are often overdispersed with respect to the Poisson distribution [46,51]. Previously, Levy et al modeled measurement noise with a distribution that allows for overdispersion and in which variance scales linearly with the mean, similar to the Poisson distribution [3]. Other studies modeled read counts with a negative binomial distribution which permits an arbitrary scaling between mean and variance [52–54].

To determine how measurement noise variance scales with the mean read count, we performed the following calibration experiment. We assembled a barcoded population of *Saccharomyces cerevisiae* with a total size of $1.6 \times 10^7$ individuals that consisted of 26 subpopulations present at six different frequencies (from $10^{-5}$ to 0.40) and sequenced it with 9-fold replication to an average depth of about $2 \times 10^5$ reads per replicate (see Materials and Methods for details).

As expected, we found that the mean of the barcode frequency estimated from sequencing data is very close to the input frequency of each lineage (Fig 4A). To test the extent of overdispersion, we plotted the variance of the measured read counts $\sigma_r^2$ against the mean $\langle r \rangle$. If the measurement process was adequately described by the Poisson distribution, we would expect to observe a linear relationship with slope 1, whereas the noise model of Levy et al predicts linear scaling with a slope greater than 1 [3]. We find that the Poisson model fits our data quite well for read counts ≤100 but fails for more abundant barcodes (Fig 4B). Instead, the variance is better fit by a non-linear convex function of the mean. We capture this nonlinearity with the model

$$\sigma_r^2 = \langle r \rangle + \epsilon \langle r \rangle^2 ,\tag{2}$$

where $\epsilon$ is a measurement noise parameter that controls the degree of overdispersion (Fig 4B, blue line). We estimate that $\epsilon = 2.4 \times 10^{-3}$ in the calibration experiment, which is significantly different from zero (S2 Table). Since the specific value of $\epsilon$ might depend on experimental details, we consider equation (2) as a constraint on our model of noise, and allow $\epsilon$ to be a free parameter at each time interval. Since the functional form (2) requires a more flexible distribution than afforded by either the Poisson or the Levy et al distributions, we model the measurement process with a negative binomial (NB) distribution (equation (5) in Materials and Methods) with mean $\langle r \rangle = nR/N$ and the variance given by equation (2).

**Bayesian filtering.** For each lineage $i$, we seek to estimate the joint belief distributions $P_{ik}^{\text{belief}}(n, s)$ of lineage size $n$ and fitness effect $s$ at each sampling time $t_k$. First, assuming that the past belief distribution $P_{ik-1}^{\text{belief}}(n, s)$ at $t_{k-1}$ and the mean fitness $\bar{s}_k$ in the time interval $(t_{k-1}, t_k)$ are known, we obtain the prior distribution $P_{ik}^{\text{prior}}(n, s; \bar{s}_k)$ at the current time $t_k$ by projecting the past belief distribution $P_{ik-1}^{\text{belief}}$ using our model of evolution. The prior distribution $P_{ik}^{\text{prior}}$ represents our best guess for the abundance of lineage $i$ and its fitness at the current time point $t_k$, based on its past trajectory and the population's mean fitness. The observed read count $r_{ik}$ for this lineage may or may not be consistent with this guess, and we use this new data to update our belief distribution using the Bayes' theorem (Fig 3B),

$$P_{ik}^{\text{belief}}(n, s) = \frac{1}{Z_{ik}^{\text{sel}}} P^{\text{meas}}\left(r_{ik}|n; \epsilon_k\right) P_{ik}^{\text{prior}}\left(n, s; \bar{s}_k\right) \text{ for } k \geq 1,\tag{3}$$

where $P^{\text{meas}}$ is the NB distribution defined in the "Model of measurement" section and $Z_{ik}^{\text{sel}}$ is the marginal probability that $r_{ik}$ reads are observed at $t_k$, given all of our prior knowledge (subscript "sel" stands for "selection" and refers to the fact that this model allows for the fitness of the lineage to be non-zero). Note that $Z_{ik}^{\text{sel}}$ depends on the noise parameter $\epsilon_k$ and the mean fitness $\bar{s}_k$ in the current time interval $(t_{k-1}, t_k)$, which are assumed to be known.

**Initialization.** We initialize the belief distribution for any lineage $i$ as $P_{i0}^{\text{belief}}(n, s) = \gamma_i(n)N(s)$, where $\gamma_i(n)$ is a Gamma distribution with parameters based on the lineage's initial read count and $N(s)$ is the normal distribution with zero mean

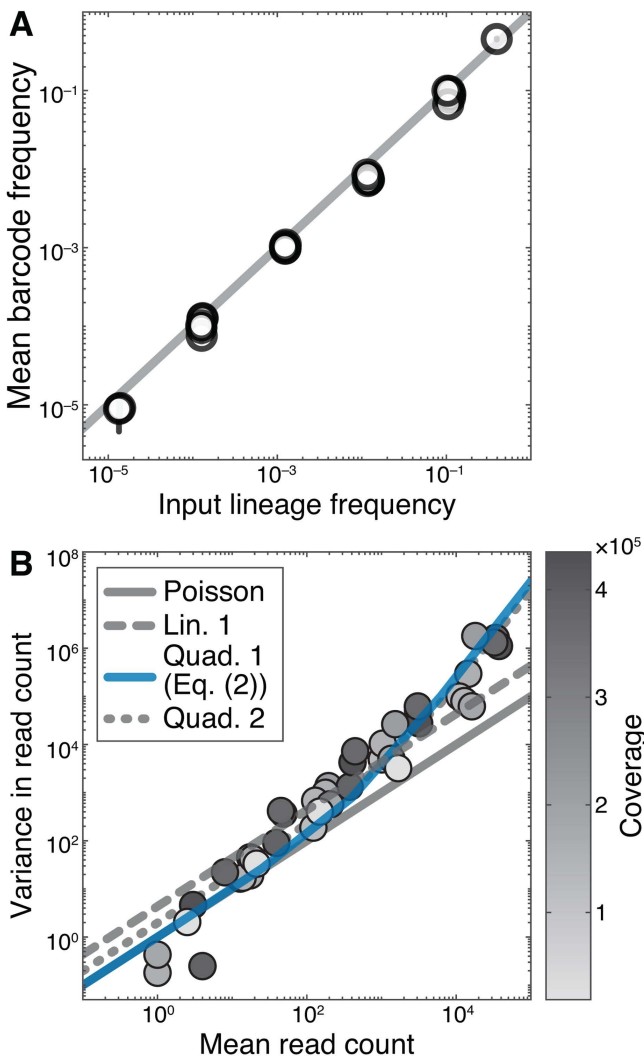

**Fig 4. Properties of the measurement process. A.** Barcode frequency estimated from barcode sequencing data (abscissa) versus the lineage frequency set experimentally (ordinate). Each point is a unique barcode. Error bars show one standard deviation of the mean. The 1-to-1 line is shown in gray. **B.** Variance of the read count versus the mean read count. Each point represents data from multiple barcodes at the same expected lineage frequency. Shade represents coverage. Solid blue line is the parabola given by equation (2). Other curves are described in the Materials and Methods and their fitted parameter values are provided in S2 Table.

and standard deviation 0.1. In other words, we assume that all lineages are initially most likely neutral, but with large uncertainty.

**Estimating population's mean fitness.** As mentioned above, our procedure for updating the belief distribution at time $t_k$ depends on the mean population fitness $\bar{s}_k$ and the noise parameter $\epsilon_k$ at the time interval $(t_{k-1}, t_k)$. To estimate $\bar{s}_k$ and $\epsilon_k$, we randomly choose 3,000 lineages and classify them as either putatively adapted (set $A_k$) or putatively neutral (set $N_k$). A lineage is classified as putatively adapted if the mean of its marginal belief distribution $P_{ik-1}^{\text{belief}}(s)$ is positive and at least three times larger than its standard deviation; and it is classified as putatively neutral otherwise. We then estimate $\bar{s}_k$ and $\epsilon_k$ by maximizing the log-likelihood function

$$L_k\left(\bar{s}, \epsilon\right) = \sum_{i \in A_k} \log Z_{ik}^{\text{sel}}\left(\bar{s}, \epsilon\right) + \sum_{i \in N_k} \log Z_{ik}^{\text{neut}}\left(\bar{s}, \epsilon\right),$$

(4)

where $Z_{ik}^{\text{neut}}$ is a quantity analogous to $Z_{ik}^{\text{sel}}$ but for the neutral model where the selection coefficient of the lineage is zero.

**Identification of adapted lineages.** As a result of the procedure described above, for each lineage $i$, we obtain a time-varying belief distribution $P_{ik}^{\text{belief}}(n, s)$, $k = 1, 2, \ldots$ We calculate the final marginal belief distribution for the selection coefficient of lineage $i$ as $P_i^{\text{belief}}(s) = \int_0^\infty P_{ik'(i)}^{\text{belief}}(n, s)dn$ choosing the time point $t_{k'(i)}$ where the $s$-variance of the belief distribution $P_{ik}^{\text{belief}}(n, s)$ is minimal. The choice to minimize $s$-variance (rather than simply using the last available time point) is motivated by the fact that secondary adaptive mutations arising later in the experiment may cause widening of the belief distribution. While such widening may be biologically informative, the main purpose of BLT experiments is to estimate the effects of single nascent adaptive mutations, suggesting that one should select a time point before secondary mutations become sufficiently abundant. In practice, this choice appears unimportant, as the $s$-variance is minimized at the last time point for the overwhelming majority of lineages across all datasets that we analyzed (see Tab C in S1 Data).

Once we obtain the lineage's final belief distribution $P_i^{\text{belief}}(s)$, we call lineage $i$ adapted if the mean $\hat{s}_i$ of this distribution is sufficiently separated from 0, that is, if $\hat{s}_i \geq \beta \sigma_i$, where $\sigma_i$ is the standard deviation of $P_i^{\text{belief}}(s)$. We refer to $\beta$ as the "confidence factor", a hyper-parameter that must be determined empirically. Intuitively, larger $\beta$ would require a lineage to have a higher fitness to be recognized as adapted, which would increase precision but reduce recall, or, equivalently, reduce the rate of false positives at the expense of increasing the rate of false negatives. In other words, a higher confidence factor would allow us to be more confident that the lineages identified as adapted are in fact adapted, albeit at the expense of missing more adapted lineages for which evidence of adaptation is weaker. On the other hand, decreasing $\beta$ would increase recall but reduce precision, or, equivalently, it would allow us to identify more adapted lineages, albeit at a cost of also misidentifying more neutral lineages as adapted. Recall and precision usually exhibit a trade-off, and there is no general principle for choosing $\beta$. In the next section, we test our method on simulated data, which provides us with a guideline for the value of the confidence factor.

## Comparison of methods on simulated data

For an initial assessment of BASIL performance and to determine the value of the confidence factor $\beta$, we applied BASIL to simulated BLT data described above (see Section "Existing BLT analysis methods can produce biased fitness estimates" and Section 4 in S1 Text). We found that BASIL accurately captures the dynamics of mean fitness in both regimes (compare Figs 1 and 5), even at relatively late time points where the other approaches fail. To determine the hyper-parameter $\beta$, we varied it between 0.0 and 6.4 with the step size of 0.1 and calculated the F1 score for adapted lineage calls, which is the harmonic mean of recall and precision (Fig 5E and 5G). We found that the F1 score is maximized at $\beta = 3.4$ and $\beta = 3.2$ for the weak and strong selection regimes, respectively. Thus, we chose the value of $\beta = 3.3$ for all our subsequent analyses (see Section 4.1 in S1 Text and S3 Fig). Using this confidence factor, we identified 2,570 adapted lineages (430 false negatives) in the weak selection regime, with 31 false positives, resulting in 98.8% precision and 85.7% recall. In the strong selection regime, we identified 2,481 adapted lineages (519 false negatives) with 7 false positives, resulting in 99.7% precision and 82.7% recall. For lineages identified as adapted, our fitness estimates $\hat{s}_i$ are both accurate and precise in both regimes. The full comparison between methods is provided in Table 1).

While the performance characteristics of our method are comparable or superior to those of the existing approaches, BASIL nevertheless fails to identify a significant proportion (between 15 and 20% in our simulations) of truly adapted lineages (Table 1). This is probably a biological inevitability rather than a flaw of the method itself. Some lineages, particularly those with weak adaptive mutations, may simply not reach large enough population sizes before they begin to decline due to clonal interference, which would prevent them from being identified as adapted by any method. Indeed, we find that false negative lineages in our simulations are typically only slightly more fit than the ancestor and their trajectories are

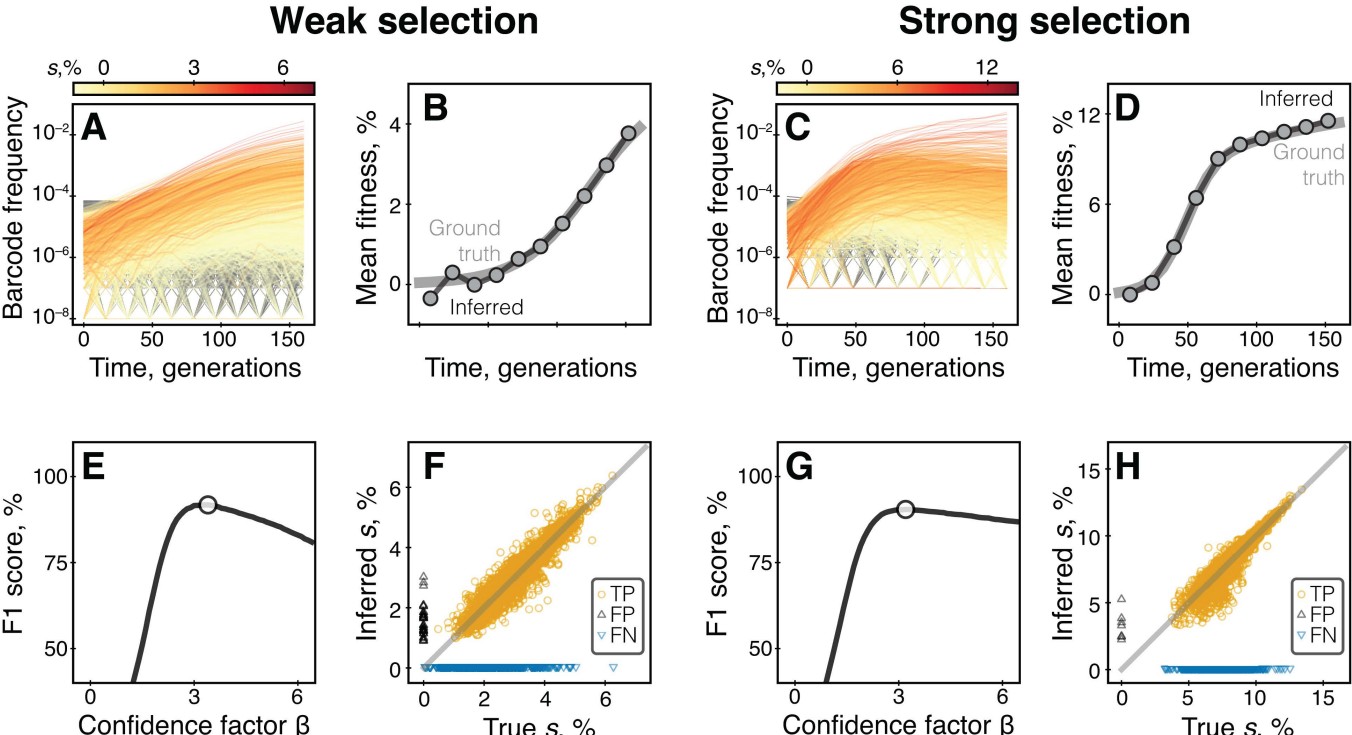

**Fig 5. Performance of BASIL on simulated data.** Left panels A, B, E, F show simulations in the weak selection regime; right panels C, D, G, H show simulations in the strong selection regime. **A, C.** Observed barcode frequency trajectories colored by fitness. **B, D.** Mean fitness trajectories. Grey lines show the ground truth, circles show values inferred by BASIL. **E, G.** F1 score as a function of the confidence factor β. White circle denotes the maximal value. **F, H.** Inferred versus true selection coefficients of individual lineages. Lineages are called adapted with β = 3.3 (see main text). Each symbol corresponds to a lineage; orange circles, black upward triangles and blue downward triangles represent true positives, false positives and false negatives, respectively; true negatives are not displayed for clarity.

**Table 1. Comparison of BLT analysis methods on simulated data.**

| Measure of performance | Weak selection | | | | Strong selection | | | |
|---|---|---|---|---|---|---|---|---|
| | Neutral Decline | | FitMut2 | BASIL | Neutral Decline | | FitMut2 | BASIL |
| | LA Ref[1] | HA Ref[2] | | β = 3.3 | LA Ref[1] | HA Ref[2] | | β = 3.3 |
| TP[3] | 2204 | 2426 | 1871 | 2570* | 1189 | 643 | 1279 | 2481* |
| FP[4] | 37 | 49 | 19* | 31 | 1 | 1 | 0* | 7 |
| FN[5] | 796 | 574 | 1129 | 430* | 1111 | 2357 | 1721 | 519* |
| Precision[6], % | 98.3 | 98.0 | 99.0* | 98.8 | 99.9 | 99.8 | 100.0* | 99.7 |
| Recall[7], % | 73.5 | 80.9 | 62.4 | 85.7* | 63.0 | 21.4 | 42.6 | 82.7* |
| F1 score[8], % | 84.1 | 88.6 | 76.5 | 91.8* | 77.3 | 35.2 | 59.8 | 90.4* |
| Error in $\bar{s}$[9], % | 0.75 | 0.19 | 0.07* | 0.12 | 4.69 | 7.07 | 3.65 | 0.16* |
| Error in $\hat{s}$[10], % | 0.64 | 0.27 | 0.40 | 0.24* | 1.29 | 7.99 | 4.41 | 0.39* |

[1]Low-abundance (LA) lineages used as reference; [2]High-abundance (HA) lineages used as reference; [3]Number of true positives; [4]Number of false positives; [5]Number of false negatives; [6]Precision = TP/(TP + FP); [7]Recall = TP/(TP + FN); [8]F1 score = 2TP/(2TP + FP + FN); [9]Absolute difference between true and inferred mean fitness, averaged across time; [10]Absolute difference between true and inferred fitness averaged over true positive lineages. *indicates the best performance.

essentially indistinguishable from those of truly neutral lineages, which supports our conjecture (see Section 4.2 in S1 Text and S4–S6 Figs for more details).

## Comparison of methods on published BLT data

We next sought to compare the performance of BASIL against FitMut2 on data from published BLT experiments. In the absence of ground truth, we took two approaches to carry out this comparison. First, Venkataram et al sampled 410 yeast clones from their BLT experiments and estimated their fitness in two separate competition assays, carried out in the presence and absence of the alga *Chlamydomonas reinhardtii* [33]. Although the fitness of these clones may differ from the fitness of the corresponding adapted lineages in the original BLT experiments, these estimates represent the best available approximation of the ground truth. Thus, we asked how accurately BASIL and FitMut2 predict the fitness of these clones based on the frequency trajectories of the corresponding lineages in the BLT experiments. Second, we evaluated how the performance of FitMut2 and BASIL degrades as we subsample the Levy et al data [3], which is the largest and the "cleanest" BLT dataset available so far (see Tab C in S1 Data and Fig 2E).

**Comparison with competition assay measurements in Venkataram 2023 data.** We find that the Pearson correlation coefficient between the fitness of yeast clones measured in competition assays and the corresponding BLT fitness estimates produced by BASIL is 0.751 and 0.636 (S7 Fig) for the experiments with and without the alga, respectively; and the average difference between the estimates is 2.56% and 3.52%. For FitMut2, the Pearson correlation coefficients in the two environments are 0.257 and 0.47 and the average difference between the estimates is almost three times larger than for BASIL, 6.02% and 10.34%, respectively.

**Down-sampling analysis on Levy 2015 data.** To test how the amount of data affects BASIL and FitMut2 performance, we down-sampled the Levy 2015 dataset [3] in two ways (see Materials and Methods). First, we reduced average coverage from about 160× to about 10× per lineage. Second, we reduced the sampling frequency from 12 time points to 8. We then compared how various performance metrics for each method degrade relative to the full dataset.

The results of this analysis are shown in Fig 6. We find that when coverage is reduced, the mean fitness trajectories inferred by either method remain virtually unchanged (Fig 6A and 6D) but the number of lineages identified as adapted decreases. Specifically, BASIL identifies 49,330 lineages as adapted in the reduced data compared to 60,920 in the full data, a 19% decrease. In contrast, FitMut2 identifies only 1,131 lineages as adapted in the reduced data compared to 15,386 in the full data, a 92.6% decrease. For both methods, the overwhelming majority (≥ 98.8%) of lineages identified as adapted in the reduced data are also identified as such in the full data, and the estimates of selection coefficients remain largely unchanged (Pearson correlation coefficient of 95.8% and 82% for BASIL and FitMut2, respectively; and an average change of 0.20% and 0.83% in the estimated value of $s$, respectively; Fig 6B and 6E and S4 Table). Notably, for BASIL, lineages that are no longer identified as adapted after down-sampling are those with small selection coefficients whereas FitMut2 loses adapted lineages even with large selection coefficients (Fig 6C and 6F and S4 Table).

When we reduce sampling frequency, BASIL exhibits a similarly modest degradation in performance. The number of adapted lineages decreases by 21% (from 60,920 to 48,206), with 99.9% calls being the same as in the full dataset. Their inferred fitness effects also remain consistent (Pearson correlation coefficient of 97.9% and 0.18% change in $s$, Fig 6H and S4 Table). One noteworthy difference compared to reduction of coverage is that, when sampling is reduced, BASIL loses power to identify lineages with both weak and strong adaptive mutations (compare panels C and I in Fig 6).

In contrast to BASIL, FitMut2 exhibits a dramatic degradation of performance under sampling frequency reduction. Pathologically, in addition to 15,386 lineages identified as adapted in the full data, FitMut2 finds 39,862 additional likely spurious adapted lineages in the reduced data, most of which have small selection coefficients (Fig 6L and S4 Table). Moreover, for those lineages that are identified as adapted in both full and reduced data, FitMut2 estimates different selection coefficients (Pearson correlation coefficient of 62.0% and 1.29% change in $s$, Fig 6K and S4 Table).

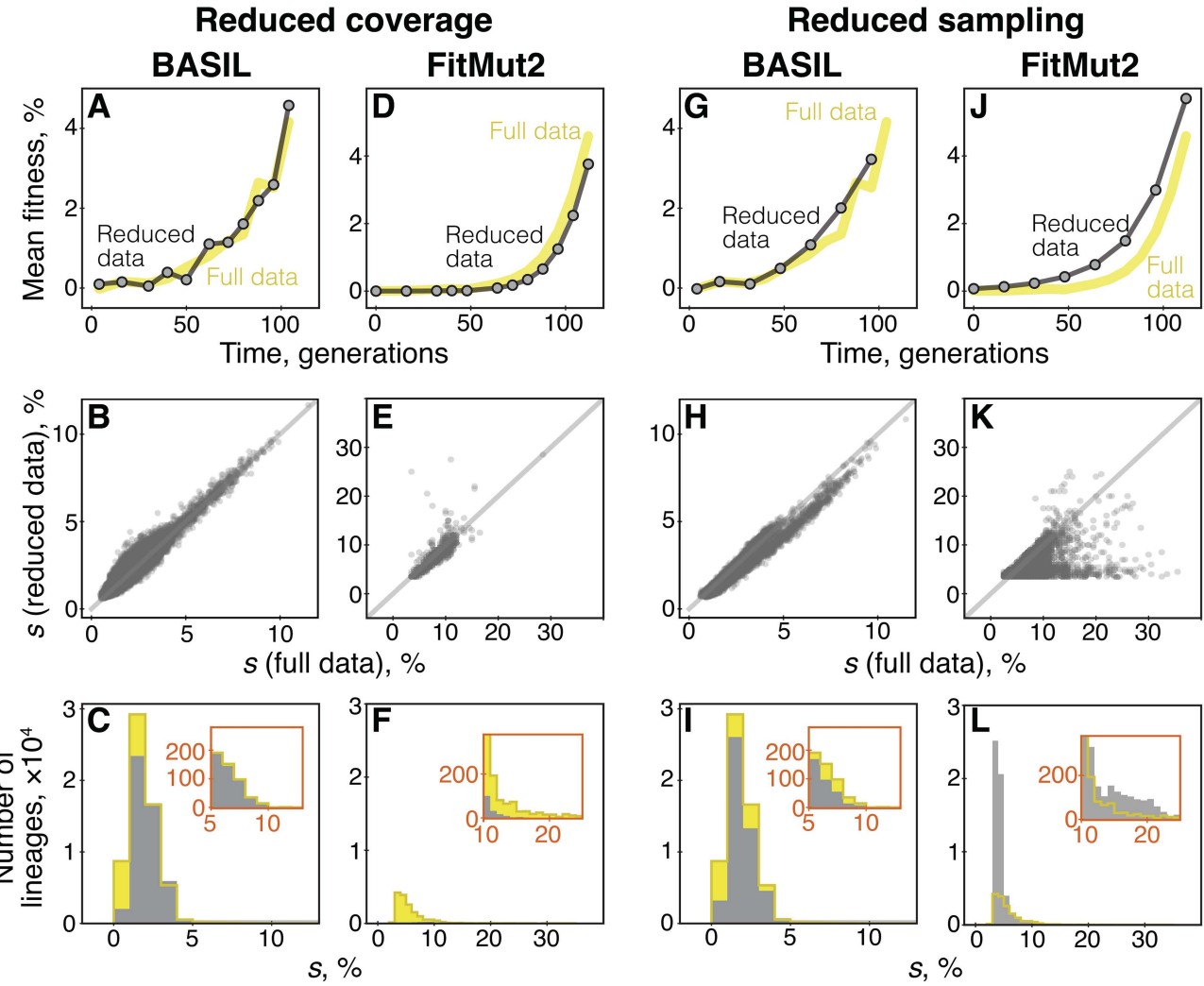

**Fig 6. The effects of down-sampling on BASIL and FitMut2 performance.** We reanalyzed the Levy 2015 dataset (Replicate 1) using BASIL and Fit-Mut2 after reducing sequencing coverage or sampling frequency. See Materials and Methods for details. **A.** Mean fitness trajectories inferred by BASIL on full and reduced data. **B.** Inferred selection coefficients for lineages identified as adapted by BASIL in both full and reduced datasets. The 1-to-1 line is shown in light gray. **C.** Distribution of fitness effects of lineages identified as adapted by BASIL in full and reduced data. Inset shows the tail of the distribution. **D–F.** Same as A–C but for FitMut2. **G–L.** Same as A–F but for reduced sampling.

In summary, reduction in sampling frequency is more detrimental for inference than a uniform reduction in coverage across all time points for both methods. Yet, while both types of data reduction degrade BASIL's performance only to moderate degree, FitMut2 exhibits a dramatic loss of power under coverage reduction and a troubling increase in the rate of false positives under sampling frequency reduction. Thus, taken together, these analyses demonstrate that BASIL is a more statistically robust, accurate and precise approach for the analysis of BLT data than FitMut2.

### Application of BASIL to data from BLT experiments

Finally, we applied BASIL to five published BLT experiments (including Levy et al [3] and Venkataram et al [33]) carried out with various strains of yeast *S. cerevisiae* propagated in different environmental conditions (see Tab C in S1 Data).

The main goal of this analysis was to expose BASIL to a variety of evolutionary scenarios, examine its outputs and look for potential failure modes that we have not observed so far. To this end, we analyzed BLT experiments described in Refs. [3,4,29,32,33].

We find that the mean fitness inferred by BASIL is generally consistent across replicates and with a posteriori estimates calculated based on the inferred selection coefficients and frequencies of identified adapted lineages (S8 Fig and Tab C in S1 Data; see Materials and Methods). However, in the Venkataram 2023 data, we find that the mean fitness inferred by BASIL begins to decline at later time points. Since there is a strong theoretical expectation that the mean fitness in large populations should not decline, this observation indicates that BASIL fails to accurately infer mean fitness at later time points in this dataset. We suspect that this failure is caused by the prevalence of secondary adaptive mutations, i.e., adaptive mutations arising in already adapted lineages [4,5]. Given that adaptation in this experiment appears to be extremely fast (0.28% per generation on average; Tab C in S1 Data), secondary adaptive mutations likely arise sooner than in other experiments. How secondary mutations can lead to declines in the inferred mean fitness can be rationalized as follows. As lineages acquire secondary adaptive mutations, their frequencies begin to rise faster than expected from previous observations. However, at later time points, BASIL has high confidence in the fitness estimates of most lineages. Thus, as it estimates the mean fitness at the next time interval, it attributes the unexpected frequency increases of multiple lineages to a decline in the mean fitness. We note that, while BASIL's apparent inability to accommodate changes in belief distributions at later stages of the BLT time course is sub-optimal, it does not represent a major problem. Indeed, the main purpose of BLT experiments is to measure the fitness effects single-mutant neighbors of a given wildtype genotype and to isolate some of them for further investigation [3,29,32,33,37,38]. Thus, the focus on typical BLT experiments is on the early stages of adaptation before secondary mutations become prevalent (but see Refs. [4,5] that examine longer-term dynamics).

Next, we examine the fitness effects of individual adaptive mutations. We find that the number of identified adapted lineages can vary across replicate BLT experiments by up to threefold (Tab C in S1 Data). We also find that, while the measured distributions of fitness effects (mDFEs) look similar across replicate experiments (Figs 7 and S9), they are statistically distinguishable by several measures (S5 Table). These differences could arise from real biological variation, measurement noise, or both. As discussed above, variation in coverage and sampling frequency leads to variation in statistical power to detect beneficial mutations (see Section "Comparison of methods on published BLT data"). Our analysis showed that this variation in power is modest in the Levy 2015 data and would not be sufficient to fully explain the differences between the numbers of detected adaptive mutations across replicates. Biologically, strong rare beneficial mutations could arise in different replicates at different times, causing real biological variation in the mean fitness trajectories (S8 Fig), which, even if small, could in turn lead to variation in the probability of establishment of weaker beneficial mutations. These observations highlight the difficulties of inferring the shape of the true underlying DFE from the fitness of detected adapted lineages.

## Discussion

In this work, we showed that the existing approaches to the analysis of BLT data have significant shortcomings. Using simulated data with pre-existing mutations, we found that both the original Levy-Blundell (LB) method and its improved version implemented in the FitMut2 package, can produce biased estimates of the fitness effects of new mutations, particularly under strong selection. We provided the likely mechanistic explanation for the suboptimal performance of the LB method. Using data from the Levy et al BLT experiment [3], we showed that FitMut2 can both exhibit significant loss of statistical power and elevated rates of false positives when the amounts of available data are reduced.

To overcome these shortcomings of existing approaches, we developed BASIL, a Bayesian filtering method for identifying barcoded lineages that carry nascent adaptive mutations and for inferring their selection coefficients. BASIL estimates belief distributions for these selection coefficients and the corresponding lineage sizes, updating them over time based on

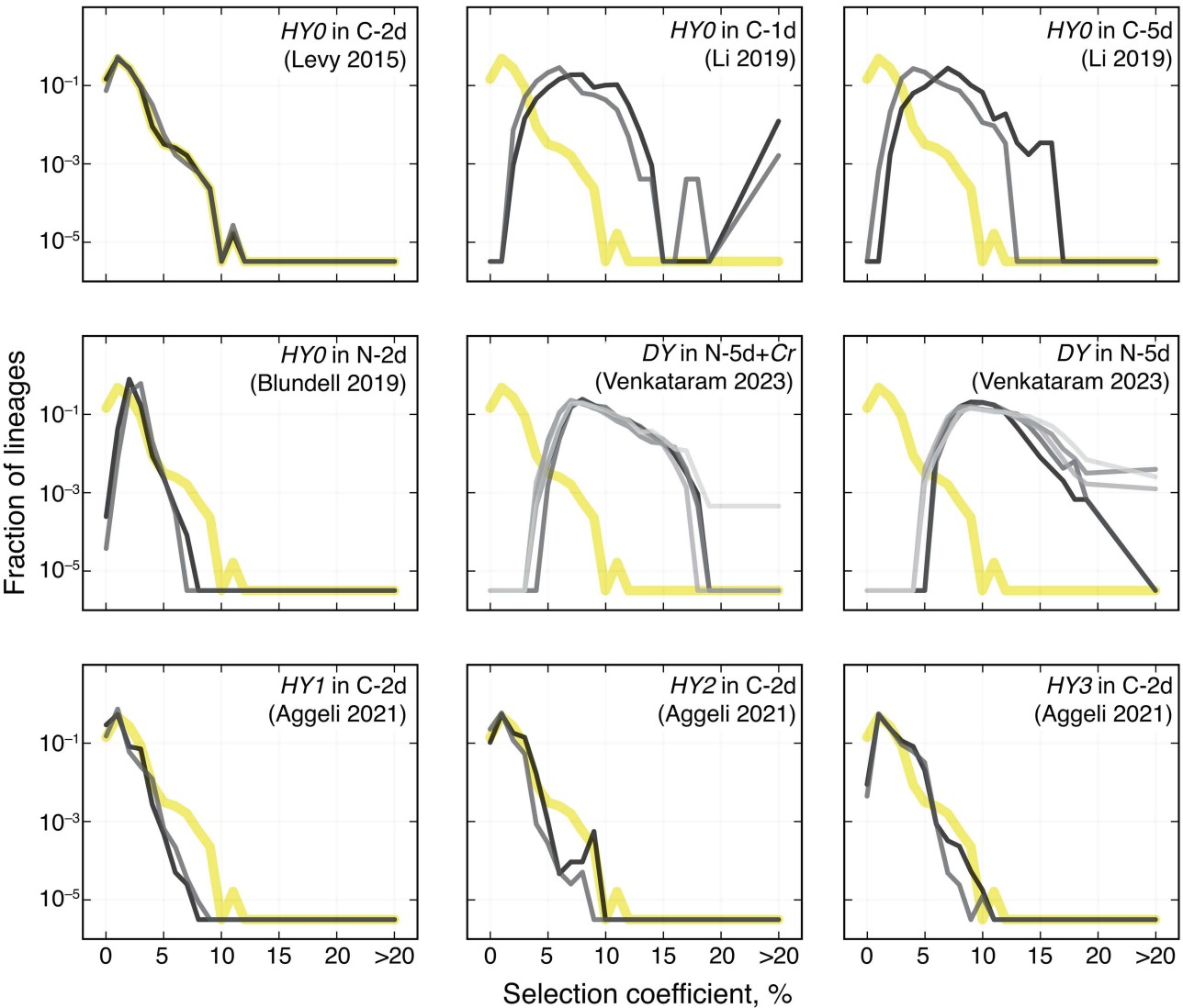

**Fig 7. Histograms of the measured distributions of fitness effects (mDFEs) of adapted lineages inferred by BASIL in nine BLT experiments.** Each panel shows a BLT experiment, as indicated. Gray lines are different replicates. In the top right corner, we provide a short-hand description of the strain (haploids are denoted HY*i*, diploids are denoted by DY) and environment used in each experiment. "C" and "N" refer to the likely limiting nutrient in the medium, -*x*d indicates the number days in the growth and dilution cycle, and "+*Cr*" indicates the presence of the alga *Chlamydomonas reinhardtii* in the evolution environment (see Tab C in S1 Data for additional details). Thick yellow line shows the mDFE of *HY0* in C-2d (Levy 2015, Replicate 1) as reference.

the observed barcode read counts. Measurement noise is a key feature of our model, and our new experimental data suggests that the measurement variance in the read count scales quadratically with the expected read count, as opposed to a linear scaling that has been postulated in previous studies. This scaling is implemented in BASIL, and we suggest to use it in future barcode-analysis models. One notable advantage of our framework is that it estimates mean fitness without any a priori designated neutral reference lineages. This makes our approach applicable to a wide range of evolutionary scenarios, including those with strong selection.

When we applied BASIL to real BLT data, we found that, in contrast to FitMut2, BASIL exhibits only a moderate loss of power when the sequencing coverage or sampling frequency are reduced, and its estimates of selection coefficients of adaptive mutations remain consistent.

BASIL computes a time-dependent joint belief distribution for the selection coefficient and size of every lineage. So far, we did not utilize all the information contained in this distribution, but instead chose to classify lineages into adapted and neutral with a simple linear classifier that is based on a snap-shot of this distribution. In our simulations, this simple classifier with the confidence factor of 3.3 (which controls the slope of the classification line) was very effective. How effective it is on real data, which is arguably more complex than our simulations, is difficult to ascertain. We visualize our classification on two-dimensional diagnostic plots where the mean of the fitness belief distribution is plotted against its standard deviation (S3 and S10 Figs). In some datasets, these plots reveal clustering of lineages, with some clusters clearly containing highly adapted lineages (S10 Fig), similar to our simulations (S3 Fig). It is possible to isolate these highly adapted clusters by adjusting the confidence factor, which can be easily done by the user in the current implementation of BASIL. However, while such an adjustment would likely improve precision, it may significantly worsen recall, since not all adapted lineages belong to the highly adapted clusters, as we saw in our simulations (S3 Fig). Instead, developing more sophisticated classifiers and testing them on more realistic simulations could be a valuable direction for future research.

Our approach has certain limitations. On the technical side, BASIL is quite computationally expensive since it requires extensive Markov Chain Monte Carlo (MCMC) sampling to estimate posterior distributions of lineage fitness and sizes. For instance, the analysis of a BLT experiment with $5 \times 10^5$ lineages sampled at 10 time points took us ~50 hours of running time on AMD Ryzen 5 7600X 6-Core Processor, and the running time increases linearly in the number of lineages and sampling time points. However, since MCMC computations for different lineages are independent, the algorithm is easily parallelizable. In our analyses, we found it convenient to use 12–32 processor cores.

One important conceptual limitation of our model is that it assigns a single constant selection coefficient to each barcoded lineage, but many barcoded sub-populations are in fact polymorphic. Indeed, each new adaptive mutation arises in a single cell, and the mutant lineage initially constitutes only a small fraction of the parental barcoded sub-population [3]. The mutant lineage outcompetes the wildtype within this sub-population gradually, such that the barcode frequency increases slower than would be expected from the mutant's selection coefficient alone. Since BASIL ignores these intra-lineages dynamics, it probably underestimates the selection coefficient of lineages that arise later in the BLT experiment and also undercounts them. However, the magnitude of this bias is hard to predict a priori because it depends on the rate of adaptive mutations and their fitness effects, the very quantities that we are trying to estimate.

A barcoded sub-population also becomes polymorphic when secondary adaptive mutations arise and begin to establish [5]. In principle, BASIL is capable of updating the belief distribution for any given lineage if its fitness changes, but in practice, convergence to a stable new distribution is probably slow at later time points when the confidence in the current belief is already high. As a result, when secondary mutations become dominant, BASIL can fail. While this represents a limitation, it is not a major one since the purpose of typical BLT experiments is to estimate the selection coefficients of single adaptive mutations rather than their combinations. Thus, when secondary beneficial mutations become common, BLT experiments become much less informative and are typically aborted. But given this limitation, BASIL is currently not expected to be reliable in BLT experiments under fluctuating or frequency-dependent selection or in spatially structured populations where expansion of adapted lineages is sub-exponential [55]. However, it should be possible to extend our approach to these more general cases.

It is important to keep in mind that BASIL only identifies adapted lineages and infers their selection coefficients but does not infer the rate at which beneficial mutations arise. To infer the latter, we would need to know not only how many beneficial mutations were detected but also how many arose but were lost by drift while rare and how many were established but not detected. In other words, we would need a model of mutant's evolutionary dynamics as well as a model of sampling and detection. Previously, Levy et al developed the former model and inferred the rates of adaptive mutations with

different fitness effects [3], but their model did not account for biases introduced by sampling and detection. Our results suggest that these biases may not be negligible (Figs 6 and 7 and S5 Table). Therefore, incorporating them into a model of beneficial mutation rate inference is an important problem for future research.

## Materials and methods

### Estimation of noise in the barcode frequency measurement process

**Strains.** 26 clones were isolated from the barcoded library of the diploid strain GSY6699 of yeast *Saccharomyces cerevisiae* used in our previous work [33]. Details of the strain and library construction are provided in Refs. [3,31]. Briefly, each clone carries a unique 30-bp DNA barcode that replaces one copy of the YBR209W locus; all clones are otherwise identical in the rest of the genome. Barcode sequences of all clones have been determined previously [33] and are reported in Tab B in S1 Data.

**Culture preparation.** To probe the measurement noise process, we aimed to construct a barcoded clone mixture consisting of 5 clones at each of five frequencies 0.1, 0.01, $10^{-3}$, $10^{-4}$, $10^{-5}$, and one clone at the remaining frequency of approximately 0.45 (26 clones total). We started a liquid culture of each clone from frozen stocks in 3 ml of YPD (10 g yeast extract, 20 g peptone, 20 g dextrose, 1 L Milli-Q water) in 16-mm test tubes, and incubated these cultures overnight, in a shaking incubator set at 200 rpm at 30°C. After overnight growth, culture densities were measured with a Coulter Counter and were found to be around $1.5 \times 10^8$ cell/ml. The measured values for all cultures are provided in Tab B in S1 Data. For clones with the four lowest target frequencies (those below and including 0.01), we diluted the overnight cultures 1:10, 1:100, $1:10^3$ and $1:10^4$ in PBS, respectively. To make the final clone mixture, we pooled 200 µl of each diluted culture, then added 200 µl of the five cultures with target frequency 0.1, and then added 600 µl of the culture with the largest frequency. This resulted in final mixture volume of 5.6 ml at density approximately $4.6 \times 10^7$ cell/ml with barcodes clustered around six distinct frequencies: 0.395, 0.108, 0.019, $1.24 \times 10^{-3}$, $1.28 \times 10^{-4}$ and $1.41 \times 10^{-5}$. Our estimates of clone frequencies in the mixture are provided in Tab B in S1 Data. We spun down the clone mixture, re-suspended it in 20 ml of PBS (final density $1.57 \times 10^7$ cell/ml, split it into 20 aliquots (1 ml each in a 2 ml cryogenic tube), supplemented with 500 µl of 80% glycerol, and stored at –70°C.

**Barcode sequencing.** We followed the same protocols for DNA extraction and sequencing library preparation as in our previous work [33]. We repeated this procedure 10 times, each time using one aliquot of the barcoded clone mixture stocks described above. All ten replicate libraries were sequenced on the Illumina HiSeq platform, resulting in approximately $2.3 \times 10^{-6}$ total reads.

**Data analysis.** We used BarcodeCounter2 to identify barcodes in the sequencing data and count them [56]. The data are shown Tab B in S1 Data. One replicate was excluded from analysis due to low coverage. In addition, we could not identify barcodes for two out of 26 clones, and two other clones (in the lowest frequency class) were not detected in the sequencing data. These four barcodes were removed from further analysis. Thus, we retained 22 barcodes present in all $n_{\text{rep}} = 9$ replicates.

Denote by $X_p$ the set of barcodes $b$ with the same target frequency $p$ and denote the read count for barcode $b$ in replicate $i$ by $r_{bi}$. For each frequency class $X_p$ and replicate $i$, we estimate the variance in the read count due to measurement noise as $\hat{\sigma}_{pi}^2 = \frac{1}{|X_p|} \sum_{b \in X_p} (r_{bi} - \bar{r}_{bi})^2$, where $\bar{r}_{bi} = \frac{R_i}{n_{\text{rep}}} \sum_{j=1}^{n_{\text{rep}}} \frac{r_{bj}}{R_j}$ is the estimate of the expected read count for barcode $b$ in replicate $i$, $R_i$ is the total coverage in replicate $i$ and $|X_p|$ denotes the total number of barcodes in the frequency class $p$. More details are provided in Section 1 in S1 Text.

We model the relationship between read count mean $\bar{r}_{bi}$ and variance $\hat{\sigma}_{pi}$ shown in Fig 4B as $\hat{\sigma}_{pi}^2 = a\bar{r}_{pi} + \epsilon\bar{r}_{pi}^2$ and test four special cases of this equation. To recapitulate the Poisson distribution, we set $a = 1$ and $\epsilon = 0$. We refer to the noise model proposed by Levy et al (see equation S8) as "Linear 1". In this model, $\epsilon = 0$ and $a$ is a free parameter. To capture the possibility that the degree of overdispersion varies across read depths, we test a quadratic relationship with $a = 1$ and a fitting parameter $\epsilon > 0$, which matches the Poisson model for small $\bar{r}_{bi}$. This is the model in equation (2), and we refer to

it as "Quadratic 1". Finally, we also fit the general quadratic model with two fitting parameters $a$ and $\epsilon$, which we refer to as "Quadratic 2".

We find the best-fit parameters using the non-linear least squares method. Since the estimated measurement noise variances span six orders of magnitude (see Fig 4B), we carry out the fitting in the log space, i.e., we minimize the squared error between $\log \widehat{\sigma}_{pi}^2$ and $\log \left( a\bar{r}_{pi} + \epsilon\bar{r}_{pi}^2 \right)$. In S1 Table we report the best-fit parameters, residual sums of squares (RSS), as well as the coefficient of determination $R^2$, computed as 1–RSS/TSS, where TSS denotes the total sum of squares. We compare the goodness-of-fit for all pairs of nested models using the $F$-test. Specifically, assuming that the null model has $p_{null}$ free parameters and the alternative model has $p_{alt}$ parameters $p_{null} < p_{alt}$, we calculate the $F$-statistic as $F = \frac{\text{RSS}_{null} - \text{RSS}_{alt}}{\text{RSS}_{alt}} \frac{n - p_{alt}}{p_{alt} - p_{null}}$, where $n$ denotes the number of data points, and $\text{RSS}_{null}$ and $\text{RSS}_{alt}$ are the residual sums of squares for the null and alternative model, respectively. If the data is generated by the null model, then the $F$-statistic is distributed according to the $F$-distribution with $(p_{alt} - p_{null}, n - p_{alt})$ degrees of freedom.

The results of this analysis are shown in Fig 4B and S1 and S2 Tables. We find that the quadratic models explain at least 93% of variance in our data compared to linear models, which explain 89% or less. We also find that the Quadratic 2 model provides only a marginally better fit than the Quadratic 1 model, with a $P$-value of only 0.0235. Therefore, we use the Quadratic 1 model (equation (2)) as our model for the relationship between read count mean and variance.

## BLT simulations

We carry our simulations in the weak and strong selection regimes (S1 Fig). We chose the parameters for the weak selection regime to make it similar to the experimental conditions described by Levy et al [3]. In particular, selection and genetic drift are relatively weak, and the variance in the read count is a linear function of the mean. For the strong selection regime, we chose parameters that should make inference harder. In particular, selection and drift are stronger, and the variance in the read count is a quadratic function of the mean (see equation (2)). Simulation parameters for both regimes are provided in S3 Table.

All simulations start with the same number $N_L = 10^5$ of barcoded lineages. The initial size of each lineage is drawn from an exponential distribution with mean $\langle N \rangle / N_L$, where $\langle N \rangle$ is the expected total population size (see S3 Table). In each simulation, 3,000 randomly chosen lineages are adapted, such that all members of such lineages have the same beneficial mutation. We draw these lineages uniformly with the constraint that their initial size must exceed $1.5D = 384$ individuals, where $D = 256$ is the dilution factor. The selection coefficient (fitness) $s_i$ of each adapted lineage is drawn from the normal distribution with mean $\langle s \rangle$ and standard deviation $\sigma_s$, and negative draws are discarded. The fitness of all other lineages is set to $s = 0$. No new mutations arise during the simulation.

After initializing our populations, we simulate their growth and dilution, with each growth-dilution cycle consisting of the following steps:

1. Each cycle begins with the calculation of the population's mean fitness $\bar{s} = \sum_i s_i n_i / N$, where $n_i$ is the current size of lineage $i$ and $N = \sum_i n_i$ is the current population size.

2. We then simulate the dilution process. For each lineage $i$, we draw its new size $n_i'$ as a random number from the Poisson distribution with mean $n_i/D$. Thus, after dilution, the population's bottleneck size equals $N_b = \sum_i n_i' \approx \langle N \rangle / D$.

3. We then simulate population expansion. Each lineage $i$ expands by the factor $W_i = D \exp \left[ (s_i - \bar{s}) \Delta t \right]$, where $\Delta t = 8$ generations is the duration of the growth phase. Thus, at the end of the growth phase, the new lineage size is $n_i^{new} = n_i' W_i$ (rounded to the nearest integer) and the total population size is again $\sum_i n_i^{new} \approx D \sum_i n_i' \left( 1 + (s_i - \bar{s}) \Delta t \right) \approx \langle N \rangle$.

Each simulation lasts for a total 20 growth-dilution cycles (160 generations). We simulate sampling and barcode sequencing by selecting $R$ random individuals from the population at the end of every other cycle (i.e., before dilution). For each sampled barcode $i$, the number of sequencing reads $r_i$ is drawn from the negative binomial distribution with mean

$\langle r_i \rangle = n_i R/N$ and variance $\sigma_r^2$, where $R$ is the sequencing depth (see S3 Table). The output of each simulation is a table of "sequenced" barcode counts at all sampling time points.

## The neutral decline method

The neutral decline method retains the essential features of the original LB method but simplifies it in two major ways. First, in the LB model, adapted lineages are characterized by two parameters, the selection coefficient $s$ and the establishment time $\tau$, whereas the neutral decline method has only one parameter $s$ per lineage. However, in our simulations, all adapted mutations are pre-existing, implying that our simpler one-parameter model is sufficient. The second important difference is that, in addition to the mean fitness $\bar{s}_k$, the LB method also estimates the noise parameter $\kappa$ (see Section 2 in S1 Text). To infer both $\bar{s}_k$ and $\kappa$, the LB method uses a model that describes how the entire distribution of read counts changes over time (equations (S8), (S9)). The mean of this distribution is described by equation (1), which does not depend on $\kappa$. Thus, equation (1) is sufficient if we are interested only in $\bar{s}_k$ (see S2 Fig). Importantly, since equation (1) directly follows from the LB model, a failure of this equation would imply a failure of the full LB model. Therefore, the focus our analysis on the neutral decline rather than the full LB model is justified.

The neutral decline method proceeds as follows.

1. For each consecutive time interval $(t_{k-1}, t_k)$, we choose the neutral reference lineages as those having certain read counts $r_{k-1}$ at the beginning of the interval with either $20 \leq r_{k-1} \leq 40$ (low-abundance reference) or $80 \leq r_{k-1} \leq 100$ (high-abundance reference).

2. For each initial read count $r_{k-1}$, we calculate the mean fitness $\bar{s}_k$ using equation (1) and then average it over all initial read counts (for reference lineages only). We repeat the mean fitness calculation for all sampling intervals and thereby obtain the full mean-fitness trajectory $\bar{\mathbf{s}} = (\bar{s}_1, \bar{s}_2, \ldots)$.

3. To infer the fitness effect $s_i$ of lineage $i$, we maximize the log-likelihood function of the lineage trajectory $r_i = (r_{i1}, r_{i2}, \ldots)$, $\hat{s}_i = \mathrm{argmax}\, \ell_i(s)$, where $\ell_i(s) = \sum_k \log p(r_{ik}; \mu_k(s, \bar{s}_k), \kappa)$ with $p(r; \mu, \kappa)$ given by equation S1 Text, and $\mu_k(s, \bar{s}_k)$ given by equation S1 Text and $\kappa = 3$.

4. To determine whether lineage $i$ is adapted, we apply the likelihood ratio test to compare the neutral model ($s_i = 0$, null hypothesis) and the adaptive model ($s_i = \hat{s}_i$, alternative hypothesis). The likelihood-ratio test statistic is given by $R_i = -2(\ell_i(0) - \ell_i(\hat{s}_i))$. We call the lineage adapted if $\hat{s}_i > 0$ and $R_i \geq 3.84$, which is the 95th percentile of the $\chi^2$-distribution with 1 degree of freedom.

## Empirical test of the neutral decline equation

In Fig 2, to test the neutral decline equation (1), we use the barcode read count data from two BLT simulations and the three BLT experiments Levy 2015 (replicate 1 in Ref. [3]), Li 2019 (replicate 2 from the 1-day transfer cycle experiment from Ref. [29]), and Venkataram 2023 (replicate 5 of the experiment with the alga from Ref. [33]). For each real or simulated BLT experiment, we selected one "early" and one "late" sampling time interval $(t_{k-1}, t_k)$, as described in the caption to Fig 2. For each interval, we calculate the average read count $\bar{r}_k$ and the rescaled average read count $\tilde{r}_k$ for lineages that had the same read count $r_{k-1} \in [0,100]$. If the number of lineages with a given $r_{k-1}$ was less than 5, we did not estimate the corresponding $\bar{r}_k$. To investigate whether $\tilde{r}_k$ is a linear function of $r_{k-1}$ as predicted by equation (1) for neutral lineages, we plot $\tilde{r}_k$ against $r_{k-1}$. For some $r_{k-1}$ in some datasets, this dependence becomes very noisy or clearly deviates from linearity. Thus, we first visually identify the intervals of $r_{k-1}$ where the data behaves approximately linearly. We choose the interval $5 \leq r_{k-1} \leq 15$ for the Li 2019, Venkataram 2023 and the strong-selection simulated datasets at the late time point; we choose the interval $5 \leq r_{k-1} \leq 40$ for the Venkataram 2023 dataset at the early time point; and we choose $5 \leq r_{k-1} \leq 99$ for all the other datasets. Then, using the data only within these intervals, we fit a linear model $\tilde{r}_k \sim a r_{k-1} + b$ either with $b = 0$ (restricted

model) or with $b$ being a free parameter (full model) and ask whether the full model provides a statistically better fit to our data using the $F$-test (see Section "Estimation of noise in the barcode frequency measurement process").

## BASIL

The detailed description of BASIL is given in Section 3 in S1 Text. Here, we provide the key equations and brief descriptions of the algorithm.

**Model of measurement.** We model the probability of observing $r$ reads for a lineage with $n$ cells in a total population of size $N$ which was sequenced to depth $R$ as

$$P^{\mathrm{meas}}\left(r|n;\epsilon\right) = \frac{\Gamma\left(r + 1/\epsilon\right)}{\Gamma(r+1)\Gamma\left(1/\epsilon\right)}\left(\frac{\epsilon\langle r\rangle}{1 + \epsilon\langle r\rangle}\right)^{r}\left(\frac{1}{1 + \epsilon\langle r\rangle}\right)^{\frac{1}{\epsilon}}$$

(5)

where $\langle r\rangle = nR/N$ is the expected number of reads.

**The belief distribution for lineage size and fitness.** Our main goal is to estimate the belief probability distributions $P_{ik}^{\mathrm{belief}}(n, s) \equiv P^{\mathrm{belief}}(n, s|\boldsymbol{r}_{ik})$, where $\boldsymbol{r}_{ik} = (r_{i0}, \ldots, r_{ik})$ is the read count vector for lineage $i$ up to and including sampling time point $t_k$. We express these belief distributions as $P^{\mathrm{belief}}\left(n, s|\boldsymbol{r}\right) = P^{\mathrm{belief}}\left(n|s, \boldsymbol{r}\right) P^{\mathrm{belief}}\left(s|\boldsymbol{r}\right)$ with parametric forms $P^{\mathrm{belief}}\left(n|s, \boldsymbol{r}\right) = \gamma\left(n; \kappa\left(\boldsymbol{r}\right), \theta\left(s, \boldsymbol{r}\right)\right)$ and $P^{\mathrm{belief}}\left(s|\boldsymbol{r}\right) = N\left(s; \mu\left(\boldsymbol{r}\right), \sigma^2(\boldsymbol{r})\right)$, where $N\left(s; \mu, \sigma^2\right)$ is the normal distribution with mean $\mu$ and variance $\sigma^2$ and $\gamma(n; \kappa, \theta)$ is the gamma distribution with the shape parameter $\kappa$ and scale parameter $\theta$ and we use the family of functions

$$\theta\left(s, \boldsymbol{r}\right) = \frac{a\left(\boldsymbol{r}\right)}{\kappa\left(\boldsymbol{r}\right)}\exp\left[\frac{b(\boldsymbol{r})\left(s - \mu\left(\boldsymbol{r}\right)\right)}{\sigma(\boldsymbol{r})}\right].$$

Thus, our belief distributions come from a family with five parameters $\mu$, $\sigma^2$, $\kappa$, $a$ and $b$ all of which are fit based on the data vector $\boldsymbol{r}$.

At $t_0$, we set $\mu = 0$, $\sigma^2 = 0.1$ and $b = 0$ for all lineages; and for lineage $i$, we set $\kappa(r_{i0}) = r_{i0}/(1 + 0.01 r_{i0})$, $a = r_{i0}N/R_0$ where $R_0$ is the coverage at the initial time point. We believe that the prior distribution should be fairly broad (i.e., uninformative), but this particular choice is likely unimportant.

**Projecting the belief distribution.** To obtain the prior distribution $P_{ik}^{\mathrm{prior}}(n, s)$ for the sampling time $t_k$, we project the belief distribution $P_{ik-1}^{\mathrm{belief}}(n, s)$ as follows. First, we obtain the 1-cycle projected probability distribution

$$P_1(n, s) = \int_0^\infty P^{\mathrm{c}}\left(n|n'; s, \bar{s}\right) P_{ik-1}^{\mathrm{belief}}(n', s)\, dn',$$

where $P^{\mathrm{c}}\left(n|n'; s, \bar{s}\right)$ is the one-cycle transition probability given by equation (S34) in S1 Text. The marginal distribution for $s$ remains unchanged during projection. The probability that the lineage goes extinct during one cycle is $\left(\frac{D}{D+\theta}\right)^{\kappa}$ where $D$ is the dilution factor and $\kappa$ and $\theta$ are the parameters of the belief distribution $P_{ik-1}^{\mathrm{belief}}(n, s)$. Conditional on lineage survival, we approximate the distribution $P_1\left(n|s, \boldsymbol{r}\right)$ by the gamma distribution with shape parameter $\kappa_1$ and scale parameter $\theta_1$, which can be calculated as functions of $s$, $\kappa$ and $\theta$ (equations (S42) and (S43)). If multiple cycles elapse between successive sampling time points, the prior probability $P_{ik}^{\mathrm{prior}}(n, s)$ is obtained by applying this procedure recursively.

**Updating the belief distribution.** The belief distribution is updated using Bayes' theorem (equation (3)). We would like to use the same analytical parametric form as above for the updated belief distribution, but since the numerator of this equation is a complex function of both $n$ and $s$, the normalization constant $Z_{ik}^{\mathrm{sel}}$ and the moments of the right-hand side of

equation (3) cannot be computed analytically. Therefore, to estimate the parameters of the updated belief distribution, we use the Markov Chain Monte Carlo (MCMC) approach described in Section 3.2.2 in S1 Text.

**Calculation of $Z_{ik}^{neut}$.** Equation (4) for the estimation of mean fitness $\bar{s}_k$ and the noise parameter $\epsilon_k$ in the time interval $(t_{k-1}, t_k)$ depends on the marginal likelihood $Z_{ik}^{neut}$ that lineage $i$ which was classified as putatively neutral in this interval has $r_{ik}$ reads at time $t_k$. We obtain $Z_{ik}^{neut}$ as follows. First, we approximate the marginal belief distribution $P_{ik-1}^{belief}(n)$ for lineage size $n$ at $t_{k-1}$ by a gamma distribution with shape parameter $\widetilde{\kappa} = \kappa \left[ (1 + \kappa)e^{b^2} - \kappa \right]^{-1}$ and scale parameter $\widetilde{\theta} = ae^{\frac{b^2}{2}}/\widetilde{\kappa}$, where $\kappa$, $a$ and $b$ are the parameters of the original distribution $P_{ik-1}^{belief}(n, s)$. We then obtain the prior distribution $P_{ik}^{prior}(n)$ under the neutral model analogously to how we obtain $P_{ik}^{prior}(n, s)$ in the model with selection. Finally, we use MCMC to estimate $Z_{ik}^{neut}$ (see Section 3.3 in S1 Text for details).

### Performance of BLT analysis methods on experimental data

**Down-sampling.** To test how sequencing coverage and sampling effort affect our mDFE inference, we down-sampled the data from Replicate 1 of the original BLT experiment with strain HY0 in the C-2d environment [3]. The original data has the average coverage of ~160× per lineage, and sampling occurred at 12 time points, at generations 0, 16, 32, 40, 48, 64, 72, 80, 88, 96, 104 and 112. To reduce coverage, for each barcode whose read count is $r_{ik}$ in the original data, we draw a random number $r'_{ik}$ from a Poisson distribution with mean $r_{ik}/16$, such that the average coverage per lineage is ~10. To reduce sampling frequency, we remove sampling time points at generations 40, 72, 88 and 104, such that the total number of samples is reduced from 12 to 8.

**Mean fitness based on inferred adapted lineages.** Since the true mean fitness in the BLT experiments is unknown, we follow Levy et al [3] and validate the mean-fitness estimates obtained by BASIL by comparing them to a posteriori mean-fitness estimates based on the inferred selection coefficients and frequencies of identified adapted lineages $\bar{s}_k^{ADP} = \sum_i \hat{s}_i \, r_{ik}/R_k$ where $\hat{s}_i$ is the inferred selection coefficient of adapted lineage $i$.

**Comparison of replicate mDFEs.** BASIL estimates of the selection coefficients of adapted lineages for all datasets shown in Fig 7 are provided in Tabs D–L in S1 Data. To construct the measured distribution of fitness effects of new mutations (mDFE) shown in Fig 7, we group lineages based on their inferred selection coefficient $\hat{s}_i$ into 1-percent bins as well as a bin of all lineages with $\hat{s}_i > 0.2$. The statistics of these mDFEs are reported in Tab C in S1 Data.

We test the consistency of mDFEs across replicates of the same BLT experiment using quantile-quantile (Q-Q) plots (S9 Fig). We find that, by this metric, most mDFEs inferred from replicate experiments are consistent with each other, although experiments HY0-C1d [29], HY2-C2d, HY3-C2d [32] and DY-N5d [33] show some discrepancies. In addition, we compare the mean, the median, the interquartile range (IQR), and the interdecile range (IDR) across replicate mDFEs using a permutation test. To this end, we create a list of selection coefficients of all lineages identified as adapted in any of the replicates and then randomly reassign each selection coefficient to each replicate $i$ while preserving the original number of selection coefficients in each replicate. We estimate each mDFE statistic of interest for each permuted replicate. Then, for each pair of replicates it in the permuted data, we calculate the absolute value of the difference between the statistic values; and we average them over all pairs to obtain our test statistic. We repeat this permutation procedure 5,000 times to obtain the null distribution for each mDFE statistic and calculate the empirical $P$-value based on this distribution. The results of this more sensitive test are reported in S5 Table.

## Supporting information

**S1 Text. Supplementary information.**
(PDF)

**S1 Fig. Simulated BLT data.**
(PDF)

**S2 Fig. Consistency of mean-fitness inference for BLT data by Levy et al.**
(PDF)

**S3 Fig. Discrimination of adapted and neutral lineages in simulated data.**
(PDF)

**S4 Fig. The frequency dynamics of lineages in the weak-selection regime, stratified by class label.**
(PDF)

**S5 Fig. The frequency dynamics of lineages in the strong-selection regime, stratified by class label.**
(PDF)

**S6 Fig. Selection coefficients and sizes of lineages identified as adapted in simulated data.**
(PDF)

**S7 Fig. Comparison of BASIL and FitMut2 performance on Venkataram et al data.**
(PDF)

**S8 Fig. Mean fitness trajectories inferred by BASIL in real BLT datasets compared to mean fitness trajectories calculated from adapted lineages.**
(PDF)

**S9 Fig. Quantile-quantile plots for the distributions of selection coefficients of lineages identified as adapted in replicate BLT experiments.**
(PDF)

**S10 Fig. Discrimination of adapted and neutral lineages in real BLT data.**
(PDF)

**S1 Table. Fitted parameters for models of the relationship between read-count mean and variance.**
(PDF)

**S2 Table. Statistical comparison of models of the relationship between read-count mean and variance.**
(PDF)

**S3 Table. Simulation parameters.**
(PDF)

**S4 Table. Effects of down-sampling on the BASIL and FitMut2 inference.**
(PDF)

**S5 Table. Differences of mDFE statistics across replicates in real BLT datasets.**
(PDF)

**S1 Data. Aggregated Data Tables.** Tab A. Contents. Tab B. Measurement noise experiment. Tab C. Characteristics of published BLT studies and some summary statistics of the BASIL analysis. Tabs D–L. BASIL results for published BLT experiments.
(XLSX)

## Acknowledgements

We thank members of the Kryazhimskiy lab for discussions and feedback. The UC San Diego Triton Computing Cluster assisted with computational work.

## Author contributions

**Conceptualization:** Huan-Yu Kuo, Sergey Kryazhimskiy.

**Funding acquisition:** Sergey Kryazhimskiy.

**Investigation:** Huan-Yu Kuo, Sergey Kryazhimskiy.

**Methodology:** Huan-Yu Kuo, Sergey Kryazhimskiy.

**Software:** Huan-Yu Kuo.

**Visualization:** Huan-Yu Kuo, Sergey Kryazhimskiy.

**Writing – original draft:** Huan-Yu Kuo, Sergey Kryazhimskiy.

**Writing – review & editing:** Huan-Yu Kuo, Sergey Kryazhimskiy.

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
