## [Decision Letter · Decision Letter 0]

10 Jul 2025

A Bayesian Filtering Method for Estimating the Fitness Effects of Nascent Adaptive Mutations

PLOS Computational Biology

Dear Dr. Kryazhimskiy,

Thank you for submitting your manuscript to PLOS Computational Biology. After careful consideration, we feel that it has merit but does not fully meet PLOS Computational Biology's publication criteria as it currently stands. Therefore, we invite you to submit a revised version of the manuscript that addresses the points raised during the review process.

Please submit your revised manuscript within 60 days Sep 09 2025 11:59PM. If you will need more time than this to complete your revisions, please reply to this message or contact the journal office at ploscompbiol@plos.org. Please include the following items when submitting your revised manuscript:

We look forward to receiving your revised manuscript.

Kind regards,

Sergei Maslov

Academic Editor

PLOS Computational Biology

Tobias Bollenbach

Section Editor

PLOS Computational Biology

**Additional Editor Comments:**

I hope the attachment with the review by the reviewer 1 is included. If not, write to me and I can send it directly. The editorial manager here is very clunky.

**Journal Requirements:**

4) We notice that your supplementary Figures, and Tables are included in the manuscript file. Please remove them and upload them with the file type 'Supporting Information'. Please ensure that each Supporting Information file has a legend listed in the manuscript after the references list.

2) If any authors received a salary from any of your funders, please state which authors and which funders..

6)Please send a completed 'Competing Interests' statement, including any COIs declared by your co-authors. If you have no competing interests to declare, please state "The authors have declared that no competing interests exist". Otherwise please declare all competing interests beginning with the statement "I have read the journal's policy and the authors of this manuscript have the following competing interests"

**Reviewers' comments:**

Reviewer's Responses to Questions

**Comments to the Authors:**

Reviewer #1: Please see the attachment.

Reviewer #2: # Summary

Kuo and Kryazhimskiy critically evaluate an existing method (neutral decline or Levy-Blundell) for inferring fitness effects of spontaneous mutants from barcoded lineage tracking data, using simulated data and previously-published experimental data (consistently applying the same method across these data sets). Besides showing how the results depend on the average strength of selection and the crucial choice of putatively neutral lineages as a reference, they find that a major limitation of this method is errors in the inferred mean fitness, which leads to errors in all individual mutant fitnesses as well. They also make an interesting point that a barcode's read count is a biased estimator of its true frequency. Then they develop a new method called BASIL which uses an iterative Bayesian algorithm to infer mutant fitness from these data sets. The major advantage of BASIL seems to be that reference lineages for the mean fitness calculation do not need to be manually chosen. They apply BASIL to the same simulated and experimental data sets and find that it generally works much better than the older methods, especially for capturing the mean fitness under both weak and strong selection.

Altogether the paper is solid. The previous literature on these methods is highly technical and often confusing in my opinion, and having a well-written paper that critically and clearly addresses these methods (besides introducing a new and better one) is valuable. I only have a few minor comments and questions (some of which are my own curiosity). I separated these into "major" and "minor" just to highlight two comments that I think are a little more substantial, but none of these should be impediments to publication.

# Major comments

1. I understand that the authors tried to keep the main text streamlined by putting a lot of details in the supplement. But this being a technical methods paper, most readers are probably going to be people who want to implement these methods themselves, and thus will want to know those details, requiring them to flip back and forth a lot between the main text and the supplement. I would therefore suggest that the authors consider just making the main text longer and more technical so at least it can be read more linearly by its main readership. For example, Secs. 4 and 5 in the supplement are critical for anyone who wants to understand this paper, so some of that material could be moved to the main text.

2. Could the authors say a little more about how to apply their method to other data sets? In particular, I am wondering what aspects of their method need to be adjusted for each new data set or experimental system and protocols, and if there are any experimental design choices that could be made in advance to accommodate them. (I realize this is not a best-practices paper for BLT experiments, but the data analysis method inevitably informs such considerations.) For example, the authors calibrate a model of how variance in read counts scales with mean read counts by sequencing a test sample with known barcode frequencies (Fig. 4). Is that something everyone needs to do first for their own system and protocols, or do they think their specific model and parameter values will hold across most cases? What about the prior distribution (with a normal distribution of fitness with standard deviation 0.1) and the hyperparameter beta? Is there some procedure we need to follow to calibrate them for our own data/systems?

# Minor comments

1. Line 9 "recently developed": I quibble with this characterization of BLT — the Levy et al. paper that started all this is over 10 years old now.

2. Line 15 "currently standard": Is the Levy-Blundell method actually that standard? I am not familiar with every study on BLT from the last 10 years, but I didn't think that everyone who does these experiments uses the same approach.

3. Lines 43-44 "this strategy preferentially captures mutations with the strongest fitness benefits because they are least likely to be lost by genetic drift or clonal interference": How do the authors reconcile this statement with the argument in the first paragraph that clonal interference means one must know the whole distribution of beneficial mutants? I understand the authors' points here, but this latter statement seems to undercut the former. That is, if the same few mutants with strongest fitness effects consistently establish in an evolution experiment, then it seems like clonal interference isn't all that important and one doesn't need to know the whole DFE, at least if one just wants to predict evolutionary dynamics. On the other hand, if an evolution experiment yields many different established mutants, then clonal interference is significant but then one can get a decent sample of beneficial mutants just from established mutants in an evolution experiment. So doesn't that mean that high-resolution lineage tracking isn't crucial in either case? 'm playing devil's advocate here, but I'd like to know what the authors think about this.

4. Lines 131-132 "Low-abundance lineages are those represented by 20 to 40 reads, which corresponds to 20 to 40 cells": Doesn't this depend on the read depth and the number of cells in the sample used for sequencing (i.e., it assumes they are approximately the same)? Is that actually true for most of these experiments?

5. Figure 1A: I noticed that the inferred mean fitness in this case is actually negative; can the authors comment on why that happens and what it means?

6. Line 170: I think it would be clearer to denote this rescaled average read count as being for the time point $k - 1$ rather than $k$ (i.e., denote as $\tilde{r}_{k - 1}$ rather than $\tilde{r}_k$), since the idea is that it estimates the expected read count at time $k - 1$ (with true value already denoted as $r_{k - 1}$, plotted in Fig. 2).

7. Figure 2E: One thing I noticed throughout this paper was how much cleaner the Levy et al. data seems to be compared to all the more recent data sets (see also Figs. 6 and 7). Do the authors know why that is?

8. Lines 224-225 "it relies on the assumption that the measured barcode frequency is an unbiased estimator of the true lineage frequency in the population, which is in general incorrect": Can the authors say more about this? There was a very interesting section on the supplement about this (Sec. 4.3.2), but I didn't follow how it is (or isn't) related to the other issues with bias in the mean fitness and the shift in expected read counts for very low reads. In general some more explanation about to what extent these effects are all caused by the same underlying assumption, or are caused by different assumptions, would be helpful. My understanding is that there are two distinct issues, both arising from the use of Eq. 1. First is the choice of putatively neutral lineages as a reference, which is potentially arbitrary and can lead to biases. Second is the point mentioned here, which is that Eq. 1 assumes that a barcode's read frequency is an unbiased estimated of its true frequency.

9. Lines 414-415 "on the basis of identified adapted lineages": What does this mean? Does it mean that some individual lineages were determined to be adapted independently of the BLT data (e.g., by isolating clones for them and performing separate competition experiments against the ancestor)?

10. Line 473 "for the inferring": Extraneous "the"

11. Supplement Sec. 2.3: This says there were 2.3e5 total reads across all 10 replicate sequencing samples (which seems quite small to me), but in the main text (line 285) it says 2e5 reads per replicate. Is one of these a typo, or do I misunderstand?

12. Supplement Sec. 2.4: The text here talks about 26 clones (unique barcodes?), but I don't understand how those were allocated across the target frequencies in the experimental setup from Sec. 2.2. That says there are supposed to be barcodes at six distinct frequencies in the test culture, so I was expecting six unique barcodes (or an integer multiple of that).

13. Supplement Sec. 2.4, four lines above Eq. S2 "has its an unknown": Extraneous "its"

14. Supplement Sec. 4.1, Eq. S6: This model from Levy et al. always struck me as overkill, and I'd like to know what the authors of this paper think. To me it seems like the stochastic birth and death during a batch growth cycle is likely to be negligible compared to the stochasticity of the dilution step, and therefore unnecessary to model stochastically (rather than just as deterministic exponential growth). Indeed, that seems to be how the authors simulate it themselves here (Sec. 3), with deterministic growth between dilutions and Poisson sampling at the dilution. Is there something I'm missing?

15. Supplement Sec. 4.3.2, line above Eq. S11 "can also be re-write": Typo

16. Supplement Sec. 4.3.2, Eq. S15: Shouldn't there be a $\int dx x$ in the numerator of the right-hand side? The left side is an expectation value of $x$, so I think there should be an average over it on the right side.

17. Supplement Sec. 5.1, fifth line: Doesn't the model assumption $\langle r \rangle = nR/N$ contradict the point the authors just made in Sec. 4.3.2, that the read frequency $\langle r \rangle/N$ is not an unbiased estimator of the actual frequency $n/N$?

18. Figure S10: Why not plot this data as scatter points of individual mutants, with inferred fitness from one replicate vs. inferred fitness from the other replicate?

19. Supplement references: There is some inconsistent formatting here (capitalization in titles, e.g., "dna," and abbreviation/capitalization of journal titles), which the journal probably won't fix themselves.

20. Finally, I appreciated the food-themed acronyms and wondered if the authors would consider the title "Putting BASIL on a BLT."

**Have the authors made all data and (if applicable) computational code underlying the findings in their manuscript fully available?**

Reviewer #1: Yes

Reviewer #2: None

PLOS authors have the option to publish the peer review history of their article (what does this mean? ). If published, this will include your full peer review and any attached files.

**Do you want your identity to be public for this peer review?** For information about this choice, including consent withdrawal, please see our Privacy Policy .

Reviewer #1: **Yes:** Mikhail Tikhonov

Reviewer #2: No

**Figure resubmission:**
---

## [Decision Letter · Decision Letter 1]

26 Jan 2026

Dear Dr Kryazhimskiy,

We are pleased to inform you that your manuscript 'Putting BASIL in a BLT: A Bayesian Filtering Method for Estimating the Fitness Effects of Nascent Adaptive Mutations' has been provisionally accepted for publication in PLOS Computational Biology.

Best regards,

Tobias Bollenbach

Section Editor

PLOS Computational Biology

Congratulations on a great paper! Please make sure to address the remaining issue pointed out by Reviewer #2.

Reviewer's Responses to Questions

**Comments to the Authors:**

Reviewer #1: The authors’ revisions addressed all my comments and concerns. Thank you & happy holidays!

Reviewer #2: The authors have made valuable revisions to this paper, and overall I support publication. However, in response to my second major comment on providing more explicit guidelines for how to apply their method to other data sets, they said they added more details on this in "Sec. 5.5 of the SI," but as far as I can tell there is no Sec. 5 in the SI. So I'm not sure where is the material they are referring to, which I do still think is important to include so that others can easily adapt the method to their own experiments. So the editor should confirm with the authors this has been included.

**Have the authors made all data and (if applicable) computational code underlying the findings in their manuscript fully available?**

Reviewer #1: Yes

Reviewer #2: Yes

PLOS authors have the option to publish the peer review history of their article (what does this mean? ). If published, this will include your full peer review and any attached files.

**Do you want your identity to be public for this peer review?** For information about this choice, including consent withdrawal, please see our Privacy Policy .

Reviewer #1: **Yes:** Mikhail Tikhonov

Reviewer #2: No

---

## [Editor Report · Acceptance letter]

PCOMPBIOL-D-25-00612R1

Putting BASIL in a BLT: A Bayesian Filtering Method for Estimating the Fitness Effects of Nascent Adaptive Mutations

Dear Dr Kryazhimskiy,

I am pleased to inform you that your manuscript has been formally accepted for publication in PLOS Computational Biology. Your manuscript is now with our production department and you will be notified of the publication date in due course.

With kind regards,

Anita Estes
